# Apical length governs computational diversity of layer 5 pyramidal neurons

**Alessandro R Galloni[1,2], Aeron Laffere[1,3], Ede Rancz[1]\***

[1]The Francis Crick Institute, London, United Kingdom; [2]University College London, London, United Kingdom; [3]Birkbeck College, University of London, London, United Kingdom

**Abstract** Anatomical similarity across the neocortex has led to the common assumption that the circuitry is modular and performs stereotyped computations. Layer 5 pyramidal neurons (L5PNs) in particular are thought to be central to cortical computation because of their extensive arborisation and nonlinear dendritic operations. Here, we demonstrate that computations associated with dendritic $Ca^{2+}$ plateaus in mouse L5PNs vary substantially between the primary and secondary visual cortices. L5PNs in the secondary visual cortex show reduced dendritic excitability and smaller propensity for burst firing. This reduced excitability is correlated with shorter apical dendrites. Using numerical modelling, we uncover a universal principle underlying the influence of apical length on dendritic backpropagation and excitability, based on a $Na^+$ channel-dependent broadening of backpropagating action potentials. In summary, we provide new insights into the modulation of dendritic excitability by apical dendrite length and show that the operational repertoire of L5PNs is not universal throughout the brain.

## Introduction

The neocortex is thought to have a modular structure composed of 'canonical circuits' performing stereotyped computations (*Harris and Shepherd, 2015*; *Markram et al., 2015*; *Miller, 2016*). Anatomical evidence supports the existence of repeating circuit architectures that display similar general features across species and brain areas (*Carlo and Stevens, 2013*; *Douglas and Martin, 2004*; *Mountcastle, 1997*). It is generally thought that these architectural motifs serve as a physical substrate to perform a small range of specific, canonical computations (*Bastos et al., 2012*; *Braganza and Beck, 2018*).

Pyramidal neurons are the main building blocks of these circuit motifs. Across brain areas and species, their biophysical attributes endow them with nonlinear properties that allow them to implement a repertoire of advanced computations at the single cell level (*Gidon et al., 2020*; *London and Häusser, 2005*; *Spruston, 2008*). Layer 5 pyramidal neurons (L5PNs) in particular provide a striking example of how dendritic properties can underlie circuit-level computations in a laminar circuit. Their dendritic nonlinearities enable signal amplification and coincidence detection of inputs – a crucial operation to integrate feedforward and feedback streams that often send projections onto separate dendritic domains. In these cells, a single backpropagating action potential (bAP), when combined with distal synaptic input, can trigger a burst of somatic action potentials. The crucial mechanism underlying this nonlinear phenomenon is the all-or-none dendritic $Ca^{2+}$ plateau (*Larkum et al., 1999a*; *Larkum et al., 1999b*).

Morphology and intrinsic properties have a profound influence on neuronal excitability. Dendritic topology and the electrical coupling between the soma and dendrites are thought to be particularly crucial for determining a neuron's integrative properties (*Mainen and Sejnowski, 1996*; *Schaefer et al., 2003*; *van Ooyen et al., 2002*; *Vetter et al., 2001*). Recent experimental work has shown that there can be substantial variation in intrinsic properties of L5PNs depending on the

**\*For correspondence:**
ede.rancz@crick.ac.uk

**Competing interests:** The authors declare that no competing interests exist.

location within a cortical area or on the species they are recorded from (*Beaulieu-Laroche et al., 2018*; *Fletcher and Williams, 2019*). However, it is often assumed that pyramidal neurons have robust enough properties across cortical areas and brain structures to support similar computations (*Bastos et al., 2012*; *Hawkins et al., 2017*; *Larkum, 2013*; *Shipp, 2016*). For instance, analogous to L5PNs, hippocampal pyramidal neurons also display dendritic $Ca^{2+}$ APs that support coincidence detection of distal and proximal inputs (*Jarsky et al., 2005*).

If L5PNs indeed have a common repertoire of operations in support of canonical computations, one would expect the same cell type in adjacent and closely related areas to exhibit the same computational repertoire. Here we have studied the bursting properties of thick-tufted layer 5 pyramidal neurons (ttL5) in mouse primary and medial secondary visual cortices (V1 and V2m). Through systematic and rigorously standardized experiments, we describe fundamentally different operation patterns linked to morphology in the two brain areas. Through computational modelling, we reveal new insights into a biophysical mechanism linking excitability and morphology, which can account for this difference. Our results question the notion of a common operational repertoire in pyramidal neurons and thus of cortical canonical computations more generally.

## Results

We made whole-cell patch clamp recordings from ttL5 neurons in V1 and V2m in acutely prepared mouse brain slices. To ensure consistency in cell type, recordings were restricted to L5PNs projecting to the lateral posterior nucleus of thalamus (n = 117), identified using retrograde labelling with cholera toxin subunit B (CTB, *Figure 1—figure supplement 1A–E*), or to neurons labelled in the Colgalt2-Cre mouse line (n = 12) known to be ttL5 PNs (*Groh et al., 2010*; *Kim et al., 2015*). We were thus able to maintain cortical area as the primary variant when comparing V1 and V2m neurons (*Figure 1—figure supplement 1F,G*).

### Thick-tufted layer 5 pyramidal neurons in V2m lack BAC firing

To reproduce the conditions required for triggering BAC firing, we stimulated synaptic inputs near the distal tuft in L1 using an extracellular electrode in conjunction with somatic stimulation through the recording electrode (*Figure 1A*). To avoid recruiting inhibitory inputs during the extracellular stimulation and create the most favourable conditions to enable BAC firing (*Pérez-Garci et al., 2006*), we added the competitive $GABA_B$ receptor antagonist CGP52432 (1 μM) to the extracellular solution. Extracellular current pulses in L1 were adjusted to evoke either a subthreshold EPSP or a single action potential at the soma. Somatic injection of a 5 ms depolarizing current pulse through the recording electrode was used to trigger single APs. In V1 neurons, combined stimulation (with the L1 input triggered at the end of the somatic pulse) could evoke a prolonged plateau potential resulting in a burst of 3 APs. Cells were considered to be BAC firing if three or more APs could be evoked following combined somatic and L1 stimulation (each evoking no more than one AP individually). We repeated these experiments in ttL5 neurons located in V2m under the same recording conditions. Upon coincident somatic AP and extracellular L1 stimulation, BAC firing was almost never observed in V2m. In summary, BAC firing was observed in approximately half the recorded V1 neurons (10/21), while neurons in V2m showed an almost total lack of BAC firing (1/18, p = 4.6*10$^{-3}$, Fisher's exact test, *Figure 1B*).

### Critical frequency ADP is diminished in V2m ttL5 neurons

To further investigate the prevalence of dendritic nonlinearities in ttL5 neurons across visual cortices, we recorded another hallmark of dendritic $Ca^{2+}$ plateaus: a prominent somatic afterdepolarization (ADP) following a high-frequency train of somatic APs, termed critical frequency ADP or cfADP (*Larkum et al., 1999a*; *Shai et al., 2015*). We recorded the somatic membrane potential from ttL5 neurons and evoked three action potentials using 3 ms pulses of somatic current injection at frequencies ranging from 50 Hz to 200 Hz in 10 Hz increments (*Figure 1C*). In V1 neurons, increasing the AP frequency above a critical frequency typically resulted in a sudden increase in the ADP (*Figure 1C*, middle). However, when recording in V2m under the same experimental conditions, there was usually no change in ADP, even at firing frequencies as high as 200 Hz (*Figure 1C*, right). To quantify this effect, we aligned the peaks of the last AP for each frequency and measured the area of the ADP difference between the 50 Hz trace and the higher frequency traces in a 20 ms

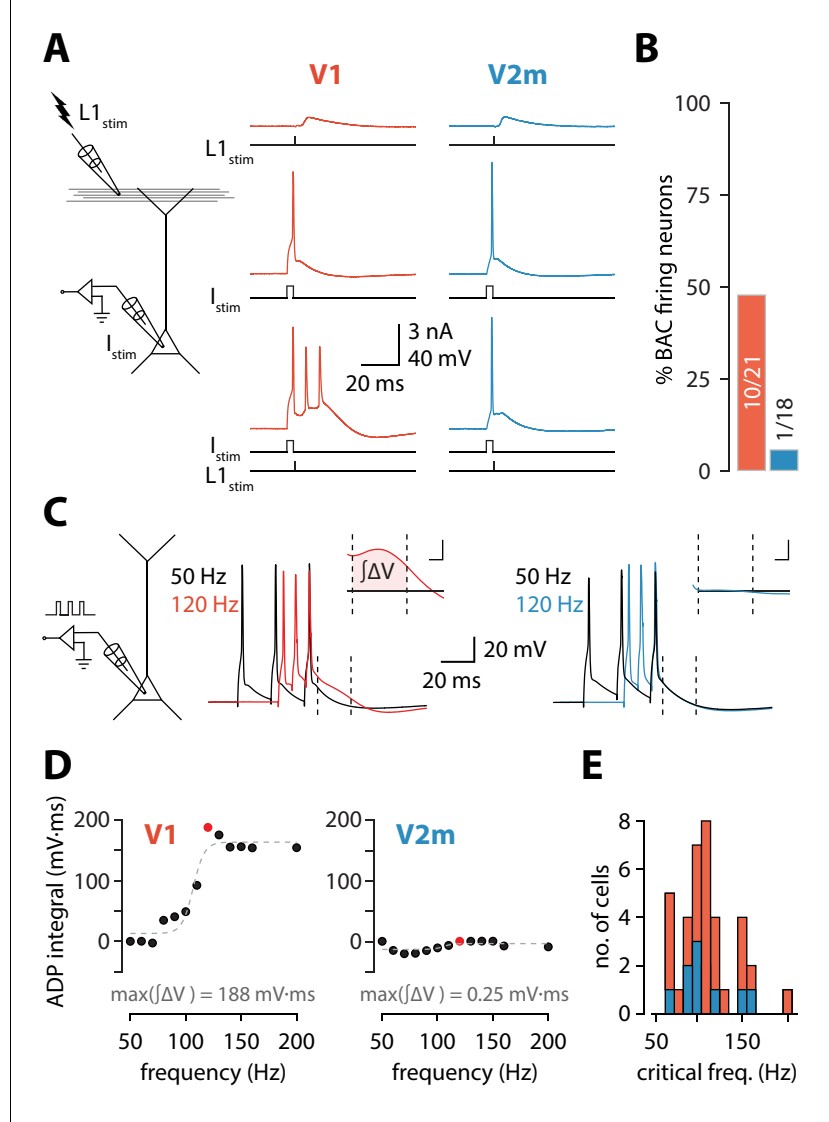

**Figure 1.** Reduced dendritic Ca$^{2+}$ electrogenesis in V2m ttL5 neurons. (**A**) *Left:* diagram of experimental configuration. *Right:* example traces during BAC firing protocol, recorded from V1 (red) and V2m (blue) ttL5 neurons. (**B**) Proportion of BAC firing neurons in V1 and V2m. (**C**) *Left:* diagram of experimental configuration. *Right:* example traces of V1 and V2m ttL5 neurons stimulated with 50 Hz and 120 Hz AP trains. Note the sustained after-depolarization following the 120 Hz spike train in the V1 neuron. *Inset:* ADP measured as the area between the 50 Hz trace and the higher frequency trace following the last spike. Inset scale bar: 5 ms x 5 mV. (**D**) Quantification of ADP area at each measured frequency for the example neurons in C. The peak integral value is highlighted in red. (**E**) Histogram of all recorded cells with defined critical frequency.

The online version of this article includes the following figure supplement(s) for figure 1:

**Figure supplement 1.** Identification of ttL5 neurons and recording locations in V1 and V2m.

window (4–24 ms) following the last AP (*Figure 1C*, inset). This measure of ADP increased sharply above a critical frequency and was often largest around the value of this frequency (*Figure 1D*). For neurons with a defined critical frequency, the mean values in V1 (112 ± 31 Hz, n = 26) and in V2m neurons (109 ± 26 Hz, n = 11) did not differ significantly (p = 0.82; two-sample t-test; *Figure 1E*).

Next, we measured the maximal ADP integral value for each cell (*Figure 2A*), regardless of the presence of a critical frequency. Neurons in V2m had significantly smaller ADP area (V1 mean = 91 ± 50 mV*ms, SD, n = 41; V2m mean = 42 ± 33 mV*ms, SD, n = 49; p = 4.54 * 10$^{-6}$, D = 0.52; two-sample Kolmogorov-Smirnov test), reflecting that most of these cells lacked a critical

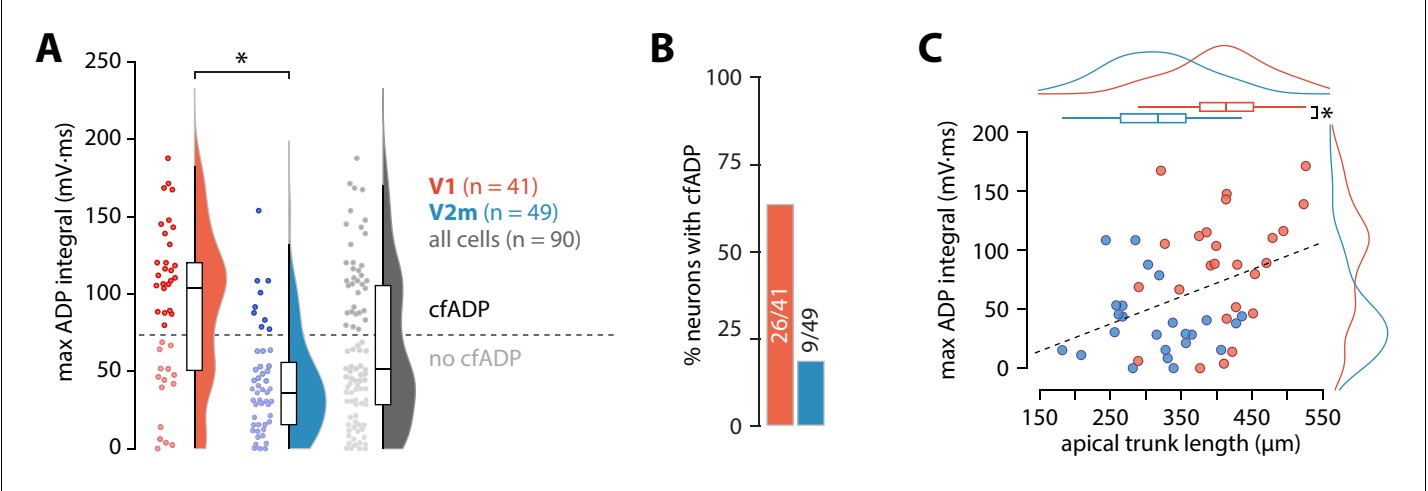

**Figure 2.** ADP integral correlates with apical trunk length. (**A**) Summary data of maximum ADP integral values for all recorded neurons. The dashed line denotes the division between the two groups of cells classified through k-means clustering; asterisk denotes p = 4.54 * $10^{-6}$, D = 0.52; two-sample Kolmogorov-Smirnov test. (**B**) Proportion of cells with cfADP in V1 and V2m. (**C**) Length of the apical trunk (soma to main bifurcation) plotted against the corresponding maximum ADP integral values. Dashed line is a linear fit ($r^2$ = 0.154, p = 5.37*$10^{-3}$, F-test); curves at the top and right are kernel density plots of the two variables in V1 and V2m; asterisk denotes p = 4.26*$10^{-6}$, two-sample t-test.

The online version of this article includes the following figure supplement(s) for figure 2:

**Figure supplement 1.** Maximum ADP integral for all cells split by recording ACSF containing either 1.5 or 2 mM CaCl$_2$.

**Figure supplement 2.** Maximum ADP integral for all cells including V1 (grey), retro-labelled cells recorded from V2m (blue), and V2m cells labelled in the Colgalt2-Cre mouse line (cyan).

**Figure supplement 3.** Intrinsic properties of in vitro and in silico recorded neurons.

**Figure supplement 4.** Reconstructions and morphological analysis of V1 and V2m neurons.

frequency altogether. As there is considerable ambiguity regarding the extracellular Ca$^{2+}$ concentration in vivo (*Lopes and Cunha, 2019*), we have used both 1.5 and 2 mM CaCl$_2$ in the artificial cerebrospinal fluid (ACSF). Although some change in the magnitude of Ca$^{2+}$ currents is expected, at the level of maximum ADP integral values there was no statistically significant difference between the two conditions in either V1 or V2m (p = 0.12, D = 0.36 for V1; p = 0.21, D = 0.29 for V2m; two-sample Kolmogorov-Smirnov test; *Figure 2—figure supplement 1*). Similarly, as there was no significant difference in the maximum ADP integral values across V2m neurons labelled retrogradely or by the Colgalt2-Cre line (p = 0.62, D = 0.24; two-sample Kolmogorov-Smirnov test, *Figure 2—figure supplement 2*), we pooled these populations. To obtain an unbiased count of cells showing cfADP, we classified the unlabelled maximum ADP values pooled from both V1 and V2m into two groups using k-means clustering with k = 2 (*Figure 2A*). The percentage of neurons with cfADP, as determined by the classification, was more than three times higher in V1 than in V2m (p = 2.6*$10^{-5}$, Fisher's exact test, *Figure 2B*).

To establish parameters which may underlie the difference in dendritic Ca$^{2+}$ electrogenesis between ttL5 neurons in V1 and V2m, we next characterized intrinsic properties of CTB labelled ttL5 neurons recorded during the cfADP experiments from V1 (n = 41) and V2m (n = 49; *Figure 2—figure supplement 3A*). The distribution of resting membrane potential, input resistance, sag amplitude, and rheobase were similar across the two populations (p > 0.05; two-sample t-test; *Figure 2—figure supplement 3D*). To capture a wide range of bursting behaviours, we devised a measure we termed the maximal burst ratio (see methods), where a high value represents strong bursting behaviour in a given neuron. In addition, we measured the number of spikes in the burst as well as the magnitude of the AHP immediately following the last spike in the burst. The maximal burst ratio, number of burst spikes, and measured AHP were significantly greater for V1 neurons than for V2m neurons (burst ratio: V1 = 12.9 ± 12.3 vs V2m = 5 ± 3.8, p = 6.1*$10^{-5}$; burst spikes: V1 = 3 ± 0.7 vs V2m = 2.3 ± 0.6, p = 6.2*$10^{-6}$; burst AHP: V1 = 17.9 ± 7 mV vs V2m = 13.6 ± 3 mV, p = 1.9*$10^{-4}$, all two-sample t-tests; *Figure 2—figure supplement 3D*). Furthermore, neurons categorized as cfADP

had significantly higher burst ratios (13.3 ± 11.9, n = 35 vs 5.7 ± 6.3, n = 55; for cfADP and non-cfADP neurons, respectively; p = 1.5*10$^{-4}$, two-sample t-test) and more prominent AHP following the burst (17.9 ± 6.9 mV, n = 35 vs 14.1 ± 3.9 mV, n = 55; for cfADP and non-cfADP neurons, respectively; p = 1.6*10$^{-3}$, two-sample t-test).

These results show that under the same conditions and in the same operational ranges V2m ttL5 neurons have similar subthreshold, yet different firing properties compared to V1. Previous research has indicated the length of the apical trunk as a possible factor involved in determining the dendritic integrative capacity of ttL5 neurons in V1 (*Fletcher and Williams, 2019*). To test this, we reconstructed the apical trunk of 22 V1 and 26 V2m neurons from those recorded. Apical trunk lengths were significantly shorter in V2m than in V1 (V1 mean = 409 ± 64 µm, SD, n = 22 vs V2m mean = 313 ± 65 µm, SD, n = 26; p = 4.26*10$^{-6}$, two-sample t-test, *Figure 2C*). Additionally, there was a correlation between maximum ADP integral values and apical trunk length across the two populations (p = 5.37*10$^{-3}$; F-test). To further test how different aspects of morphology may affect our results, we created full morphological reconstructions of 6 V1 and 7 V2m recorded neurons (*Figure 2—figure supplement 4A*). Dendritic length and number of branchpoints for the tuft and basal dendrites as well as the number of oblique branches were similar between the two populations. Apical length, convex hull and largest Sholl radius, however, were significantly greater in the V1 neurons (p < 0.005 Bonferroni-corrected two sample t-test, *Figure 2—figure supplement 4B,C*). While dendritic signals typically attenuate with distance, these results suggest that there may be a counter-intuitive interaction between apical trunk length and dendritic Ca$^{2+}$ electrogenesis in ttL5s – the further bAPs need to travel along the trunk, the more they can trigger Ca$^{2+}$ plateaus.

## BAC firing is absent in models of ttL5 neurons with short morphology

To investigate possible mechanisms underlying the dependence of bursting on apical trunk length, we ran numerical simulations in conductance-based compartmental models of ttL5 neurons. We first probed BAC firing in a morphologically detailed model developed by *Hay et al., 2011*, using the model parameters (biophysical model 3) and morphology (cell #1) favoured for reproducing BAC firing. As in the original paper, BAC firing was triggered by a 0.5 nA EPSC-like injection at the apical bifurcation coupled to a somatic action potential evoked by square-pulse current injection at the soma. Mirroring the responses seen in the subset of strongly bursting ttL5 neurons, coincident stimulation triggered BAC firing in the detailed model (*Figure 3A*, left). We then applied the same model parameters to an example V2m morphology with a shorter apical dendrite. The high density ('hot') Ca$^{2+}$ zone was shortened to 100 µm to reflect the 50% reduction in apical trunk length and repositioned around the new apical branch point (350–450 µm from the soma vs 685–885 µm in the long morphology). Intrinsic properties of this shorter model remained within the physiological range (as well as within the range of V2m recordings; *Figure 2—figure supplement 3D*). The amplitude of the dendritic current injection in the shorter model (0.217 nA) was chosen to obtain the same EPSP amplitude (~6.2 mV) at the bifurcation in both model cells. With this short morphology from V2m, coincident tuft and somatic stimulation evoked only a single somatic spike and did not trigger a dendritic Ca$^{2+}$ plateau (*Figure 3A*, right). To explore the sensitivity of Ca$^{2+}$ plateaus to dendritic Ca$^{2+}$ channel density in the long and short neurons, we scaled the total density of Ca$^{2+}$ channels (g$_{Ca}$ henceforth) to between 0 and 8 times their original quantities, while keeping the ratio of the two Ca$^{2+}$ conductances (high voltage- and low voltage activated; HVA and LVA respectively) constant. The integral of the dendritic voltage at the bifurcation, acting as an indicator of the large and sustained depolarization during a Ca$^{2+}$ plateau, increased proportionally to g$_{Ca}$ in the long morphology. In the short morphology, however, the voltage integral stayed low across all g$_{Ca}$ values (*Figure 3B*), consistent with the absence of Ca$^{2+}$ plateaus. The underlying Ca$^{2+}$ currents, while scaling with g$_{Ca}$, were 4 orders of magnitude smaller in the model with short morphology (*Figure 3—figure supplement 1A*). To explore if Ca$^{2+}$ plateaus were at all possible in the short neuron model, we systematically changed both the dendritic current injection (range: 0.02–0.5 nA, the value used in the long morphology) and the time difference between the somatic and dendritic stimulus (range:± 20 ms). The current injection threshold of dendritic Ca$^{2+}$ electrogenesis showed an asymmetric U-shaped relationship in the long morphology, as expected (*Figure 3—figure supplement 1B*; *Larkum et al., 1999b*). In the short morphology, Ca$^{2+}$ plateaus could only be evoked by the largest coincident current injections (0.5 nA, ± 1 ms). While during the largest dendritic injection the evoked dendritic potential was substantially larger than in the long morphology, it resulted in only a small

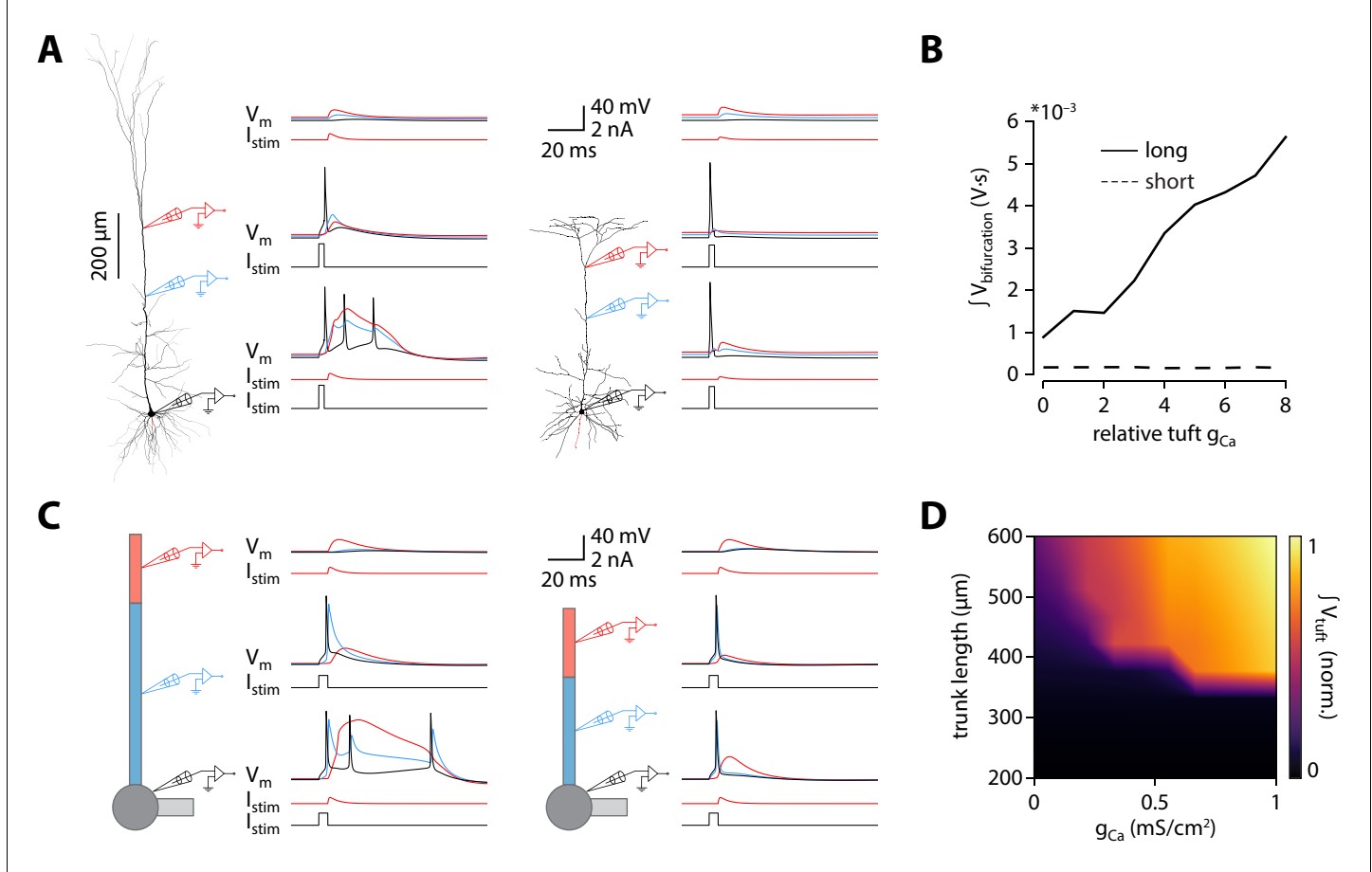

**Figure 3.** Shorter model neurons are less prone to burst. (**A**) *Left:* detailed morphology of a ttL5 neuron from the model favoured by *Hay et al., 2011* for reproducing BAC firing. *Right:* reconstructed morphology of the ttL5 neuron from V2m. Injected current and recorded voltage traces are shown for the soma (black), the apical trunk (blue, 400 and 200 µm from the soma), and the main bifurcation (red, 620 and 370 µm from the soma) under three different stimulation paradigms. (**B**) Integral of voltage at the branch point during coincident somatic and branch point stimulation, plotted against relative $g_{Ca}$. (**C**) *Left:* diagram of the reduced neuron model. Apical trunk length 600 µm. Injected current and recorded voltage traces as in A. *Right:* Same for a version of the reduced model modified to have an apical trunk length of 200 µm. (**D**) Heat map representing the normalised tuft voltage integral during combined somatic and tuft stimulation in the reduced model, plotted against the absolute density of $Ca^{2+}$ channels in the tuft compartment and the length of the apical trunk compartment. Default $g_{Ca} \cong 0.45$ mS/cm$^2$.

The online version of this article includes the following figure supplement(s) for figure 3:

**Figure supplement 1.** Exploration of $Ca^{2+}$ electrogenesis under diverse stimulus timings and intensities.

**Figure supplement 2.** Hot zone size or $Ca^{2+}$ channel ratio does not account for differences in $Ca^{2+}$ electrogenesis.

depolarization at the soma and no somatic spike burst was triggered when combined with a somatic spike (*Figure 3—figure supplement 1C*). Next we tested if the largest coincident dendritic injection (0.5 nA, ± 1 ms) could evoke BAC firing in the short morphology when combined with increased $Ca^{2+}$ conductance. While the dendritic voltage integral scaled similarly to the long morphology, showing active contribution of $Ca^{2+}$ currents (*Figure 3—figure supplement 1D*), the short neuron model never exhibited BAC firing. To further ensure comparability between the models with long and short morphology, dendritic electrogenesis was probed across a broad range of hot zone sizes (range: 50–200 µm) and different HVA:LVA ratios (range: 100:1 – 1:10). Neither the length of the hot zone (*Figure 3—figure supplement 2C,F*) nor the HVA:LVA ratio (*Figure 3—figure supplement 2D*) had a substantial effect on the difference between long and short morphologies in dendritic $Ca^{2+}$ electrogenesis, BAC firing, or on somatic ADP measures (*Figure 3—figure supplement 2*). These results indicate that, although the size of the $Ca^{2+}$ plateau depends on $g_{Ca}$, and to some

extent on hot zone length and on the ratio of HVA and LVA channels, these factors alone are not sufficient to enable BAC or critical frequency firing in short neurons.

Next we set out to explore the effect of altered apical trunk length in isolation from complex interactions found in biological morphologies. Dendritic arbours can be usefully reduced to a small set of equivalent compartments where spatial detail is abstracted away while the essential electrotonic divisions of the neuron are maintained. We selected an existing reduction (*Bahl et al., 2012*) of the *Hay et al., 2011* model for use in all subsequent in silico experiments. While altering apical trunk length in the reduced model, conductances were redistributed with the same decay constant to account for the change in apical length (see Materials and methods). Intrinsic properties remained within the physiological range as well as within the range of V1 and V2m recordings following these changes (*Figure 2—figure supplement 3D*). As with the morphologically detailed model, the reduced model (with the originally published parameters) displayed BAC firing triggered by coincident tuft and somatic stimulation (*Figure 3C*, left). Shortening the apical trunk was sufficient to eliminate this response (*Figure 3C*, right). As with the detailed model, the current injection threshold of dendritic $Ca^{2+}$ electrogenesis at different inter-stimulus intervals showed an asymmetric U-shaped relationship that diminished with shorter trunk lengths (*Figure 3—figure supplement 1E*).

We explored the dependence of BAC firing on apical trunk length and $g_{Ca}$ by measuring the time-integral of tuft voltage as an indicator of $Ca^{2+}$ plateau potentials (*Figure 3D*). The presence of a $Ca^{2+}$ plateau depended strongly on apical trunk length and was only sensitive to $g_{Ca}$ above a critical length of approximately 350 µm ($\cong 0.35\ \lambda$). Below this length, no $Ca^{2+}$ plateaus were triggered regardless of how high $g_{Ca}$ was set to. These experiments show that a reduced model can also reproduce our results, allowing us to explore and dissect the underlying parameters in more detail.

## Active propagation enhances voltage in long dendrites

To obtain a better understanding of what causes the length dependence of bursting, we made recordings from the final segment of the apical trunk as well as the tuft using the reduced model of *Figure 3C*. To recreate the experimental conditions of *Figure 1C*, we triggered 3 spikes at 100 Hz through somatic current injection. As with coincident bAP and tuft input, increasing the length of the apical trunk facilitated dendritic $Ca^{2+}$ plateau initiation (*Figure 4A*). Interestingly, the width and peak voltage in the tuft increased steadily with dendritic length (*Figure 4B,C*), even in the absence of $Ca^{2+}$ conductances ($g_{Ca} = 0$). In the presence of voltage-gated $Ca^{2+}$ channels, the increased amplitude of bAPs triggered a large all-or-none $Ca^{2+}$ plateau above a certain threshold length.

We found that bAP amplitude in the tuft increased as a function of apical trunk length despite a decreasing bAP amplitude in the distal segment of the trunk (*Figure 4B*). Conversely, the width of the bAP (measured 2 mV above baseline to capture the long tail in the voltage response without contamination by spiking events) increased in both the tuft and trunk with length (*Figure 4C*). While waveform broadening is a natural consequence of passive filtering along dendrites, the sustained voltage in the distal trunk required active dendritic propagation. In the reduced model, this active propagation in the apical trunk was mediated primarily by voltage-gated $Na^+$ channels. Removing these channels caused a substantial reduction in peak voltage and width of the depolarization in the distal trunk, and importantly also abolished the trend of increasing tuft voltages with longer dendritic trunks (*Figure 4D,E*). More generally, active propagation caused bAPs to be larger and broader at all distances along a long dendrite compared to the same absolute distances in shorter dendrites (*Figure 4—figure supplement 1A,B*). Consequently, when comparing the final positions along the trunk, the peak voltage is only marginally smaller in long dendrites despite the larger distance from the soma. This is not the case in a passive dendrite, where voltage attenuation depends almost exclusively on distance and not trunk length (*Figure 4—figure supplement 1C*), bar the end-effect in the last segment. We next tested how the specific distribution of active conductances affected the results. When all the various conductances were uniformly distributed along the apical trunk, the waveforms did not substantially change, and the enhanced voltage continued to trigger $Ca^+$ plateaus only in neurons with long apical trunks (*Figure 4—figure supplement 2*). We have also tested the specific contribution of the H-current ($I_h$). Reducing $I_h$ (i.e. $g_{HCN}$) either in the tuft alone, or both in the trunk and tuft resulted in a slower increase of tuft depolarization, width and voltage integral with apical trunk length (*Figure 4—figure supplement 3A and B*). Crucially, tuft peak voltage remained positively correlated with apical trunk length even after the total removal of $I_h$ (i.e. had positive slope), contrary to changes observed with varying the amount of sodium current in the

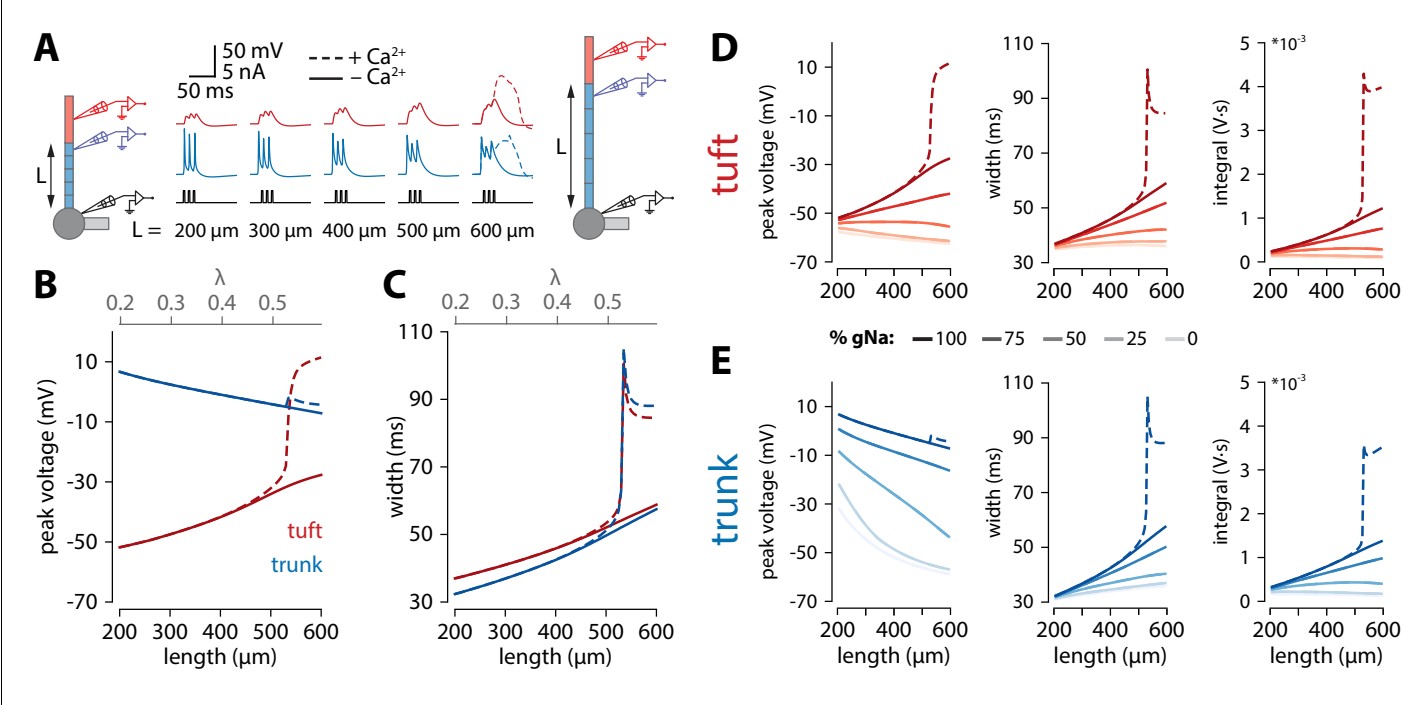

**Figure 4.** Tuft voltage increases with trunk length due to the widening of bAPs. (**A**) Schematic of the simulation: stimulation site in the somatic compartment and recording sites at the distal end of the apical trunk (blue) and in the centre of the tuft (red). Stimulus shown in black. Solid lines: $g_{Ca} = 0$; dashed lines: original $g_{Ca}$. (**B**) Peak voltage reached in trunk (blue) and tuft (red) for a range of simulations with different trunk lengths. Length constant $\lambda = 1009$ µm. (**C**) Same as in B but plotting the width of the depolarization measured 2 mV above baseline. (**D, E**) Peak voltage, width, and integral values measured in the trunk and tuft for dendrites containing different $Na^+$ channel densities in the apical trunk.

The online version of this article includes the following figure supplement(s) for figure 4:

**Figure supplement 1.** Backpropagation of APs in active and passive trunks of different length.

**Figure supplement 2.** Tuft voltage increases with trunk length independently of conductance gradients.

**Figure supplement 3.** $I_h$ is not critical to the length dependence of $Ca^{2+}$ electrogenesis.

**Figure supplement 4.** Effect of axial resistance on voltage propagation.

apical trunk (*Figure 4—figure supplement 3C*). Furthermore, dendritic $Ca^{2+}$ electrogenesis remained dependent on apical trunk length, albeit with the actual threshold length tracking the changes in $I_h$ (*Figure 4—figure supplement 3D*). It has previously been suggested that axial resistance ($R_a$) in the apical dendrite may influence the backpropagation efficiency in dendrites and burstiness of ttL5 neurons (*Fletcher and Williams, 2019*). To test this hypothesis, we measured peak voltage and width in the trunk and tuft for different trunk lengths under different $R_a$ conditions. We found that peak tuft voltage increased with increasing trunk $R_a$, reaching the highest voltage near the reduced model's original value, and decreasing again for higher values (*Figure 4—figure supplement 4*). However, in these simulations both width and voltage integral increased monotonically with trunk length regardless of the specific value of $R_a$ in both the tuft and the last segment of the trunk. This indicates that, although important, $R_a$ is not the primary determinant for generating the length-dependent effect and if $R_a$ indeed correlates with length, these effects may combine to further enhance the tuft voltage in long neurons. These results show that the general phenomenon of enhanced voltage propagation in longer dendrites resulting in amplification of tuft voltage does not depend on any of the particular model parameters above.

While it might seem counterintuitive that peak tuft voltage is increasing when trunk voltage is decreasing with length, we propose that the temporal broadening of the depolarization can at least partially account for this via a passive mechanism. Wider depolarizations allow the tuft compartment to charge to a higher voltage. The rate and peak value of tuft charging depends on the passive properties of the tuft. The peak value of depolarization and the rate of voltage change are proportional to membrane resistance ($R_m$) and membrane capacitance ($C_m$), respectively. The product of

these two parameters gives the membrane time constant ($\tau$) which determines the maximal amplitude reachable over a certain time. To illustrate this, we applied voltage-clamp to the end of the distal segment of the trunk and delivered 30 mV square voltage pulses of increasing width while recording the voltage in the tuft. Due to membrane capacitance, a short (10 ms) voltage step did not fully charge the tuft while a longer, 40 ms voltage step allowed tuft voltage to reach the steady-state value commanded by $R_m$ (*Figure 5Ai*). To directly test the hypothesis that the relationship between trunk depolarization width and tuft membrane time constant caused the amplitude of the tuft depolarization to increase with length, one could vary $R_m$ by changing $g_{leak}$. However, this would affect resting membrane potential and consequently alter voltage-dependent properties in the tuft. In order not to affect other variables in the model, we therefore chose to vary $C_m$ instead (*Figure 5Aii*). For a given value of depolarization amplitude and width, increasing tuft $C_m$ (and therefore $\tau$) caused a reduction in the tuft peak depolarization (*Figure 5B*). These simulations show that the tuft time constant and the width of the invading voltage pulse (e.g. bAP) interact to create a higher tuft depolarization with longer apical trunks.

Overall, it is the combination of increased width and a relatively small reduction in amplitude through active backpropagation that resulted in a trunk voltage integral that increased with trunk length, thereby passing more charge to the adjacent tuft compartment. When active backpropagation was reduced or absent, the trunk integral and resulting tuft voltage decreased with length (*Figure 4D,E*). The voltage integral in the distal trunk thus seemed to approximately determine the peak tuft voltage. To test this, we applied voltage-clamp to the end of the trunk and injected square

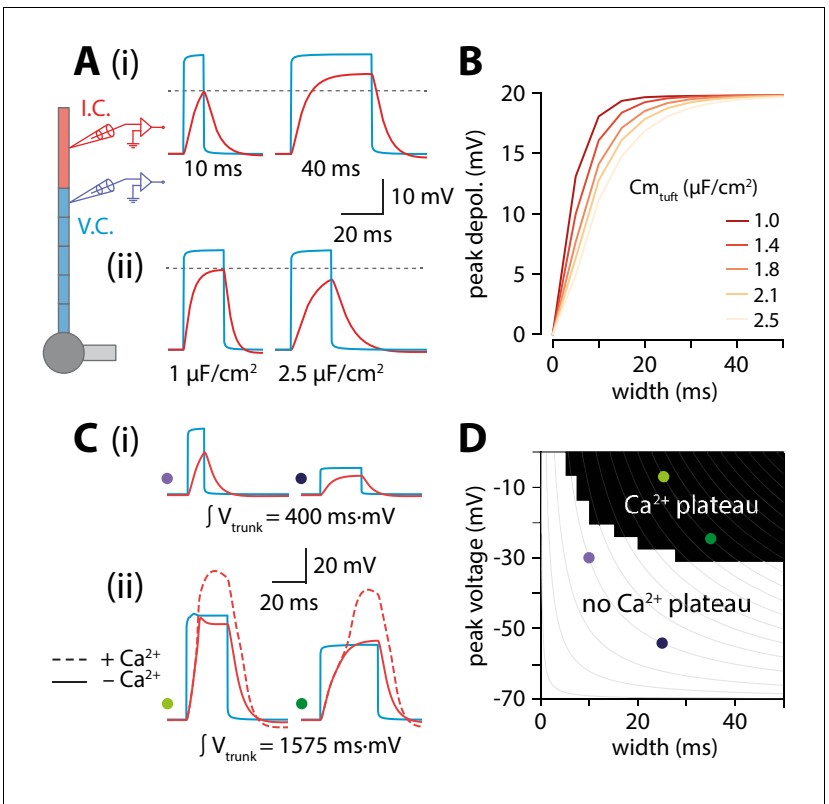

**Figure 5.** Interplay of width and amplitude underlie length dependence of $Ca^{2+}$ electrogenesis. (**A**) Different width voltage steps (i) or same width under differing tuft membrane capacitance conditions (ii) injected into the trunk (blue). Recorded tuft voltage in red, $V_{rest}$ = −65 mV. (**B**) Peak voltage values reached in the tuft for a range of trunk step widths and tuft capacitances. In the original model, tuft $C_m \cong 1.75\ \mu F/cm^2$. (**C**) Same as in A but showing voltage steps of different width and amplitude with the same voltage integral in the trunk. For smaller integrals (i) the voltage remains subthreshold while for larger integrals (ii), the tuft voltage crosses the threshold for a $Ca^{2+}$ plateau. (**D**) Voltage and width combinations for square voltage steps in the distal trunk which result in a $Ca^{2+}$ plateau in the tuft. Coloured dots represent the value combinations illustrated in C. Grey lines indicate width and amplitude combinations with equal integral.

steps with a range of integrals obtained through various combinations of width and amplitude (*Figure 5C*). This revealed a zone above a critical trunk integral for which many different width and depolarization combinations were sufficient to evoke a $Ca^{2+}$ plateau in the tuft (*Figure 5D*). At the boundary of this $Ca^{2+}$ plateau-evoking zone, iso-integral lines crossed asymmetrically into subthreshold combinations of width and depolarization. These results show that it is the complex interplay of width and amplitude with the tuft time constant that enables dendritic $Ca^{2+}$ electrogenesis.

## Discussion

We made whole-cell patch-clamp recordings from L5PNs in primary and secondary visual cortices of mice. We found that both BAC firing and cfADP were almost entirely absent in the V2m neurons. Moreover, we observed that the propensity for $Ca^{2+}$ electrogenesis was positively correlated with the length of the apical dendrite trunk across all neurons. To investigate the influence of apical trunk length on burstiness, we ran numerical simulations in compartmental biophysical models. Both morphologically detailed and reduced models showed that decreasing the apical trunk length resulted in reduced propensity for dendritic $Ca^{2+}$ electrogenesis. Further simulations revealed that this is due to an interplay between bAP width, amplitude, and tuft impedance that depends critically on the presence of voltage-gated $Na^+$ channels in the apical trunk. These results show that the same cell type, in closely related and adjacent cortical areas, and under the same operating conditions can have a very different computational repertoire. Our findings are thus inconsistent with the notion of canonical computations at the single cell level, suggesting they may not exist at the circuit level either.

Contrary to common assumptions, we observed considerable differences in the properties of ttL5 neurons across different brain regions. BAC firing, a dendritic operation, and cfADP, a measure of dendritic excitability, which are both critically dependent on dendritic $Ca^{2+}$ plateaus, were almost completely absent in V2m. One key factor known to control dendritic $Ca^{2+}$ electrogenesis is inhibition (*Pérez-Garci et al., 2006*). To determine the role of inhibition on the observed differences between V1 and V2m neurons, we pharmacologically blocked $GABA_B$ receptors, thus creating more favourable conditions for BAC firing. To make comparisons rigorous, we have used the same experimental protocols and conditions across all experiments. Furthermore, to exclude bias in selecting ttL5 neurons, we have recorded only from two well-defined groups, identified either by their projection target or genetic label. We found similarly diminished propensity for bursting in both groups of V2m neurons.

The extracellular stimulation used to evoke BAC firing could in principle recruit different polysynaptic circuits in V1 and V2m, which could account for the difference. However, this would not explain the differences measured with the cfADP paradigm using intracellular stimulation only. Another confounding variable affecting excitability could be the $Ca^{2+}$ concentration in the extracellular solution. We have found the same results under different physiologically relevant concentrations. Altogether, these results show that decreased propensity of V2m ttL5 neurons for dendritic $Ca^{2+}$ electrogenesis is a robust phenomenon.

Correlated with the differences in dendritic $Ca^{2+}$ electrogenesis, we found ttL5 neurons in V2m to have significantly shorter apical trunks compared to V1 neurons. This data is consistent with recent structural MRI data showing a thinner cortical mantle in more medial and posterior parts of the cortex (*Fletcher and Williams, 2019*). Using an existing, prominent biophysical model designed to reproduce ttL5 properties such as BAC firing, we found that the same model applied to a shorter morphology resulted in a loss of BAC firing, independently of $Ca^{2+}$ channel density. This was also true in a reduced ttL5 model with a simplified morphological structure, which allowed for continuous exploration of apical trunk length. We found a sharp cut-off at a length of 0.35 $\lambda$ ($\cong$ 350 µm in model space), below which no BAC firing could be evoked. We note, however, that the reduced model is based on rat neurons and the apical trunk and oblique dendrites are pooled into the same compartment. It is also worth noting that, as the reduced model does not have a distinct compartment to represent the apical bifurcation, all $Ca^{2+}$ channels are placed in the tuft compartment. Therefore, the numerical values of model length do not translate directly into apical trunk lengths for real mouse neurons.

Through our simulations, we identified voltage-gated $Na^+$ channels in the apical dendrite as a key factor for reproducing our results. Sodium channels control dendritic excitability by supporting active

backpropagation, resulting in reduced attenuation over distance. Combined with a broadening of the bAP proportionally to trunk length (due to capacitive filtering), this caused longer neurons to have larger voltage integrals in the distal trunk, leading to greater charging of the tuft. Indeed, the peak tuft voltage depended on the amount of passive charging it experienced, which was determined by the membrane time constant in the tuft as well as by bAP width and amplitude. We hypothesised that, above a minimal threshold for peak trunk voltage, the primary determinant of peak tuft voltage is the time-averaged voltage in the trunk. Supporting this view, we found that the overall voltage integral was more important for triggering $Ca^{2+}$ plateaus than the particular combination of depolarization width and amplitude.

Aside from dendritic length, our results do not exclude the involvement of other mechanisms to modulate excitability. For example, differences in axial resistance could influence the way voltage propagates along dendrites. While axial resistance indeed had a marked effect on the backpropagation of action potentials in our models, this was independent of trunk length and thus cannot account for our finding. Variations in density and activation properties of other voltage-gated channels may also influence dendritic excitability. It is interesting to note that, in the presence of $Na^+$ channels, the bAP in neurons with longer trunks was larger and broader at every distance from the soma. This may be due to a cooperative effect of each trunk section on the sections both up- and downstream, with the voltage at each location decaying slower because of the more depolarised state of the remaining dendrite. Our data thus predicts that bAP width and amplitude measured at the same absolute distance from the soma will be larger in neurons with longer apical trunks. It is worth noting that the models we used lacked NMDA channels. Experimental evidence shows that NMDA spikes can significantly increase the ability of distal tuft input to trigger $Ca^{2+}$ plateaus at the bifurcation. However, when NMDA spikes are triggered directly at the hot zone, they have very little contribution to $Ca^{2+}$ plateaus (*Larkum et al., 2009*). In addition, the cfADP protocol is independent of synaptically released glutamate, and hence we expect the lack of NMDA channels in the models to have no significant effect on dendritic $Ca^{2+}$ plateaus.

There are a few notable counterexamples to the principle observed here. For example, the human ttL5 neuron was recently shown to have greater compartmentalization and reduced excitability compared to rat neurons despite being substantially longer (*Beaulieu-Laroche et al., 2018*). This may still be consistent with our predictions, as human ttL5 neurons also had reduced ion channel densities, which we show to be crucial for the length-dependent enhancement. Furthermore, as the boosting effect of a broader depolarization is subject to saturation (when the depolarization is wide enough to fully charge the tuft), we would expect the positive effect of length on tuft voltage to not increase monotonically. Consequently, beyond a certain length the trunk voltage would attenuate to the point where it is no longer sufficient to trigger a $Ca^{2+}$ plateau. On the other end of the spectrum, layer 2/3 pyramidal neurons in rat barrel cortex have shorter apical trunks yet do show cfADP, although they do not exhibit spike bursts during BAC firing (*Larkum et al., 2007*). It remains to be seen if the length-dependence of dendritic $Ca^{2+}$ electrogenesis generalizes to other species, or across more widespread cortical areas and cell types.

There may be important clinical implications to gaining a better understanding of how variations in cortical thickness, and the resulting changes in neuronal morphology, affect the physiology and computational properties of pyramidal neurons. Indeed, altered cortical thickness has been implicated in several debilitating neurological diseases and mental health conditions. For example, cortical thinning is used as a biomarker for Alzheimer's disease (*Dickerson et al., 2009*) and correlates strongly with bipolar disorder (*Hanford et al., 2016*), while increased cortical thickness is present during development in individuals with autism (*Khundrakpam et al., 2017*). Interestingly, altered dendritic excitability has also been strongly implicated in several of these diseased conditions (*Hall et al., 2015*; *Nanou and Catterall, 2018*; *Spratt et al., 2019*). We uncovered a possible mechanistic link between cortical thickness and excitability, highlighting a new potential avenue of study for understanding the pathophysiology in these conditions and raising the prospect of identifying intervention targets.

Our findings on dendritic $Ca^{2+}$ electrogenesis in ttL5 neurons have wide-ranging implications for cortical computation. Feedback connectivity between cortical areas tends to target superficial layers while feedforward input favours basal dendrites (*Coogan and Burkhalter, 1990*; *Rockland and Pandya, 1979*). BAC firing is believed to play a major role in integrating these two pathways to modulate sensory perception (*Bachmann, 2015*; *Takahashi et al., 2016*) and to enable brain-wide

learning algorithms that would otherwise be intractable (*Guerguiev et al., 2017*; *Sacramento et al., 2018*). This may be even more relevant in higher order cortical areas, which are more likely to process brain-wide feedback to integrate convergent multisensory, motor and cognitive signals (*Freedman and Ibos, 2018*). However, we show that, at least within the secondary visual cortex, different operations must be implementing the multimodal coincidence detection proposed to underlie various visual phenomena (*Bachmann, 2015*).

Our findings thus challenge the commonly held notion that the neocortex is composed of canonical circuits performing stereotyped computations on different sets of inputs across the brain. The heterogeneity of operating modes may expand the ability of cortical areas to specialize in the computations that are required for processing their particular set of inputs, at the cost of reduced flexibility in generalizing to other types of input. It may also imply that the nonlinear operations performed through $Ca^{2+}$ plateaus and BAC firing may not be required outside primary sensory cortices, perhaps because in these regions the input hierarchy is less defined. It may therefore be more important to maintain equal weighting between different sensory modalities and rely on other mechanisms to change the weights according to the reliability of each input. With such simple morphological adjustments capable of generating a wide range of possible operations, the brain undoubtedly leverages the array of available computations to improve cognitive processing.

# Materials and methods

## Key resources table

| Reagent type (species) or resource | Designation | Source or reference | Identifiers | Additional information |
|---|---|---|---|---|
| Strain, strain background (*Mus musculus*, C57BL/6, male) | *Colgalt2-Cre* | The Jackson Laboratory | RRID:MMRRC_036504-UCD | |
| Strain, strain background (*Mus musculus*, C57BL/6) | *Rbp4-Cre* | The Jackson Laboratory | RRID:MMRRC_031125-UCD | |
| Strain, strain background (*Mus musculus*, C57BL/6) | Ai14 | The Jackson Laboratory | RRID:IMSR_JAX:007908 | |
| Peptide, recombinant protein | Alexa Fluor 488-conjugated Cholera toxin subunit B (CTB) | Thermo Fisher Scientific | Cat.# C34775 | 0.8% w/v |
| Chemical compound, drug | CGP-52432 | Tocris | Cat.# 1246 | 1 µM in ACSF |
| Peptide, recombinant protein | DyLight 594-conjugated streptavidin | Thermo Fisher Scientific | Cat.# 21842 | 2 µg/ml |
| Peptide, recombinant protein | Biocytin hydrochloride | Sigma-Aldrich | Cat.# B1758 | 0.5% w/v |
| Other | DAPI stain | Sigma-Aldrich | Cat.# D9542 | 5 µg/ml |
| Software, algorithm | MATLAB R2018a | MathWorks | RRID:SCR_001622 | |
| Software, algorithm | Python 3.6 | | RRID:SCR_008394 | |
| Software, algorithm | SciPy | | RRID:SCR_008058 | |

*Continued on next page*

*Continued*

| Reagent type (species) or resource | Designation | Source or reference | Identifiers | Additional information |
|---|---|---|---|---|
| Software, algorithm | NEURON simulation environment 7.7.1 | *Carnevale and Hines, 2006* | RRID:SCR_005393 | |
| Software, algorithm | Neurolucida 360 | MBF Bioscience | RRID:SCR_001775 | |
| Software, algorithm | Igor Pro 6.37 | WaveMetrics | RRID:SCR_000325 | |
| Software, algorithm | NeuroMatic 2.5 | *Rothman and Silver, 2018* | RRID:SCR_004186 | |
| Software, algorithm | FIJI | https://imagej.net/Fiji | RRID:SCR_002285 | |

## Animals

All animal experiments were prospectively approved by the local ethics panel of the Francis Crick Institute (previously National Institute for Medical Research) and the UK Home Office under the Animals (Scientific Procedures) Act 1986 (PPL: 70/8935). Transgenic male mice were used; Tg(*Colgalt2*-Cre)NF107Gsat (RRID:MMRRC_036504-UCD, referred to as Colgalt2-Cre (also known as Glt25d2-Cre); n = 11 animals) and Tg(*Rbp4*-Cre)KL100Gsat (RRID:MMRRC_031125-UCD; n = 8 animals) lines were created through the Gensat project (*Gerfen et al., 2013*; *Groh et al., 2010*) and crossed with the Ai14 reporter line expressing tdTomato (RRID:IMSR_JAX:007908). As only male mice are transgenic in the Colgalt2-Cre line, all experiments were done on male animals. Animals were housed in individually ventilated cages under a normal 12 hr light/dark cycle.

## Surgical procedures

Surgeries were performed on mice aged 3–7 weeks using aseptic technique under isoflurane (2–4%) anaesthesia. Following induction of anaesthesia, animals were subcutaneously injected with a mixture of meloxicam (2 mg/kg) and buprenorphine (0.1 mg/kg). During surgery, the animals were head-fixed in a stereotactic frame and a small hole (0.5–0.7 mm) was drilled in the bone above the injection site. Alexa Fluor 488-conjugated Cholera toxin subunit B (CTB, 0.8% w/v, Invitrogen) was injected using a glass pipette with a Nanoject II (Drummond Scientific) delivery system at a rate of 0.4 nL/s. Injections of 100–200 nL were targeted to the lateral posterior (LP) thalamic nucleus, with stereotactic coordinates: 2.2–2.5 mm posterior to bregma, 1.45 lateral of the sagittal suture, 2.45 mm deep from the cortical surface. To reduce backflow, the pipette was left in the brain approximately 5 min after completion of the injection before being slowly retracted.

## Slice preparation

Male mice (6–12 weeks old) were deeply anaesthetised with isoflurane and decapitated. In mice that were injected with CTB, this occurred at least 3 weeks after the injection. The brain was rapidly removed and placed in oxygenated ice-cold slicing ACSF containing (in mM): 125 sucrose, 62.5 NaCl, 2.5 KCl, 1.25 $NaH_2PO_4$, 26 $NaHCO_3$, 2 $MgCl_2$, 1 $CaCl_2$, 25 dextrose; osmolarity 340–350 mOsm. Coronal slices (300 μm thick) containing visual cortex were prepared using a vibrating blade microtome (Leica VT1200S or Campden 7000smz-2). Slices were immediately transferred to a submerged holding chamber with regular ACSF containing (in mM): 125 NaCl, 2.5 KCl, 1.25 $NaH_2PO_4$, 26 $NaHCO_3$, 1 $MgCl_2$, 1.5 or 2 $CaCl_2$, 25 dextrose; osmolarity 308–312 mOsm. The holding chamber was held in a water bath at 35°C for the first 30–60 min after slicing and was kept at room temperature (22°C) for the remaining time (up to 12 hr) after that. All solutions and chambers were continuously bubbled with carbogen (95% $O_2$/5% $CO_2$).

## Electrophysiology

After the 35°C incubation period, individual slices were transferred from the holding chamber to the recording chamber, where they were perfused at a rate of ~6 mL/min with regular ACSF (see above)

continuously bubbled with carbogen and heated to 35 ± 1°C. Borosilicate thick-walled glass recording electrodes (3–6 MΩ) were filled with intracellular solution containing (in mM): 115 CH$_3$KO$_3$S, 5 NaCl, 3 MgCl$_2$, 10 HEPES, 0.05 EGTA, 3 Na$_2$ATP, 0.4 NaGTP, 5 K$_2$-phosphocreatine, 0.5% w/v biocytin hydrochloride, 50 µM Alexa Fluor 488 hydrazide; osmolarity 290–295 mOsm; pH 7.3. To restrict recordings to ttL5 neurons, visually guided whole-cell patch-clamp recordings were made from neurons in layer 5 of medial V2 (V2m) and V1 that were fluorescently labelled with either CTB (n = 117 neurons) or with tdTomato (for Colgalt2-Cre mice, n = 12 neurons). Visual areas were defined based on approximate stereotaxic coordinates (*Franklin and Paxinos, 2007*; see *Figure 1—figure supplement 1*). All recordings were made in current-clamp mode. Extracellular monopolar stimulation was achieved by passing a DC current pulse (0.1–1 ms, 20–320 µA) through a glass patch-clamp pipette with a broken tip (~20 µm diameter) using a constant current stimulator (Digitimer DS3). Current was passed between two silver/silver chloride (Ag/AgCl) wires: one inside the pipette, which was filled with recording ACSF, and the other coiled around the outside of the pipette. In experiments using extracellular stimulation, 1 µM CGP52432 was added to the ACSF.

## Immunohistochemistry and morphological reconstructions

After recording, slices were fixed overnight at 4°C in a 4% formaldehyde solution and were subsequently kept in PBS. For immunohistochemical detection, the fixed slices were first incubated for 1–2 hr at room temperature in blocking solution containing 0.5% Triton X-100% and 5% Normal Goat Serum (NGS) in PBS. Slices were then washed twice (10 min each) in PBS and incubated overnight in a staining solution containing 0.05% Triton X-100, 0.5% NGS, DyLight 594-conjugated streptavidin (2 µg/ml). Slices were then washed in PBS (3 times, 5 min each) and stained with DAPI (5 µg/ml) for 10 min. After another wash (3 times, 5 min each), slices were mounted on glass slides and images were acquired with a confocal microscope (Leica SP5, objective: 20x/0.7NA or 10x/0.4NA, pinhole size: 1 airy unit). The images were used to reconstruct the apical dendrites with Neurolucida 360 (MBF bioscience). For the detailed morphological analysis, a subset of neurons, selected based on the quality and completeness of staining, was reconstructed in full through the LMtrace service of https://ariadne.ai/lmtrace.

## Data acquisition and analysis

Recorded signals were amplified and low-pass filtered through an 8 kHz Bessel filter using a Multi-Clamp 700B amplifier (Molecular Devices). Filtered signals were then digitized at 20 kHz with a National Instruments DAQ board (PCIe-6323). Acquisition and stimulus protocols were generated in Igor Pro with the Neuromatic software package (*Rothman and Silver, 2018*). Further analysis and data visualization were performed with custom macros and scripts written in Igor Pro and MATLAB (MathWorks). Confocal images were processed with Fiji (https://fiji.sc/). Classification of cfADP data in *Figure 2* and related *Figure 2—figure supplements 2* and *3* was done using the built-in MATLAB function kmeans(X,k) with k = 2. Raincloud plots (consisting of a scatter plot, a box plot, and a gaussian kernel density plot) were generated in MATLAB using published scripts (*Allen et al., 2019*). All box plots presented show the median, interquartile range, 2nd and 98th percentile of the dataset.

Sag amplitude was calculated as the difference between peak and steady-state voltage during a current step leading to ~10 mV hyperpolarization. As bursting is typically defined based on the instantaneous firing rate, which is the reciprocal of the inter-spike interval (ISI), we devised a measure, termed burst ratio, as the largest ratio of two consecutive ISIs in any given current step. Because the strongest burst occurs at different current steps for different cells, we report the maximal burst ratio across all current steps. A long ISI preceded by a relatively short one indicates that the firing rate has instantaneously gone from high to low (as occurs at the end of a burst). Thus, the higher the maximal burst ratio, the stronger the burst. This ISI-based bursting measure allowed us to reliably quantify burst strength independently of other related factors such as the number of spikes in the burst or the amount of current required to elicit a burst.

## Modelling

Simulations were performed with the NEURON simulation environment (7.7.1, *Carnevale and Hines, 2006*) embedded in Python 3.6. To model the consequences of morphological differences between V1 and V2m ttL5 neurons, we used existing models of ttL5 pyramidal cells with either accurate

morphological detail (biophysical model 3, cell #1 from *Hay et al., 2011*), referred to as detailed model) or simplified multicompartment morphologies ($Ca^{2+}$ enriched model 2 from *Bahl et al., 2012*, referred to as reduced model). To study the effect of morphology in the detailed model, biophysical model 3 from *Hay et al., 2011* was applied to the reconstructed morphology of an example ttL5 neuron in V2m (which has a substantially shorter apical trunk than the morphology used in the original model). Each morphology contained a hot zone in a 100–200 µm long region around the main apical bifurcation, with 100 times higher low-voltage-activated (LVA) and 10 times higher high-voltage-activated (HVA) $Ca^{2+}$ channel densities (same as in *Hay et al., 2011*).

Subsequent simulations using the reduced model were done by modifying only selected parameters described in the results, such as the length of the apical trunk compartment, leaving all other parameters unchanged. Briefly, the reduced model (*Bahl et al., 2012*) is divided into sections representing the soma, axon (hillock and initial segment, AIS), basal dendrites, apical trunk, and apical tuft. Active conductances are present in all compartments and include the following: hyperpolarization-activated cation (HCN) channels (basal dendrite, apical trunk, tuft), transient voltage-activated $Na^+$ (Nat) channels (soma, axon hillock, AIS, apical trunk, tuft), persistent voltage-activated $Na^+$ (Nap) channels (soma), fast voltage-activated $K^+$ (Kfast) channels (soma, apical trunk, tuft), slow voltage-activated $K^+$ (Kslow) channels (soma, apical trunk, tuft), muscarinic $K^+$ (Km) channels (soma), slow $Ca^{2+}$ (Cas) channels (tuft), calcium dependent $K^+$ (KCa) channels (tuft), and a calcium pump (tuft). The density of the Kfast and Kslow channels in each compartment decays exponentially from the soma to the tuft. The density of Nat channels decays linearly in each compartment from the soma to the tuft, while HCN channels linearly increase in density. N.B. the tuft, but not the trunk, contains $Ca^{2+}$ channels; consequently, there is no hot zone similar to the apical bifurcation in the detailed model. When varying trunk length, Nat, Kfast and Kslow, conductances in each trunk compartment were redistributed by the same algorithm as in the original model, but taking into account the new distance of each compartment from the soma (thereby changing the total conductance in the trunk). Since $g_{HCN}$ for each trunk compartment is determined in the model by the trunk length and tuft $g_{HCN}$, when varying tuft $I_h$ alone we calculate trunk $g_{HCN}$ for each length as though tuft $g_{HCN}$ were at 100%. When varying tuft and trunk $I_h$ together, we allow trunk $g_{HCN}$ to derive from the scaled tuft $g_{HCN}$ and apply no further scaling to trunk $g_{HCN}$.

All data, analysis scripts and simulation code are available at https://github.com/ranczlab/Galloni.etal.2020 (*Laffere, 2020*; copy archived at https://github.com/elifesciences-publications/Galloni.etal.2020).

## Acknowledgements

We thank Arnd Roth and Alexandra Tran-Van-Minh for advice on modelling. We are grateful to Lee Fletcher, Arnd Roth, Alexandra Tran-Van-Minh, Michael Hausser and Anna Cappellini for helpful comments on the manuscript.

## Additional information

### Funding

| Funder | Grant reference number | Author |
| --- | --- | --- |
| Wellcome | 104285/B/14/Z | Ede Rancz |
| Royal Society | 104285/B/14/Z | Ede Rancz |
| Boehringer Ingelheim Fonds | PhD Scholarship | Alessandro R Galloni |
| Francis Crick Institute | PhD Scholarhip | Alessandro R Galloni |

The funders had no role in study design, data collection and interpretation, or the decision to submit the work for publication.

### Author contributions

Alessandro R Galloni, Conceptualization, Formal analysis, Investigation, Writing - original draft, Writing - review and editing; Aeron Laffere, Conceptualization, Software, Formal analysis, Investigation,

Numerical modeling; Ede Rancz, Conceptualization, Supervision, Funding acquisition, Investigation, Methodology, Writing - original draft, Project administration, Writing - review and editing

### Author ORCIDs
Alessandro R Galloni (ID) https://orcid.org/0000-0001-9602-4274
Aeron Laffere (ID) http://orcid.org/0000-0001-8307-0079
Ede Rancz (ID) https://orcid.org/0000-0002-7951-1385

### Ethics
Animal experimentation: All animal experiments were prospectively approved by the local ethics panel of the Francis Crick Institute (previously National Institute for Medical Research) and the UK Home Office under the Animals (Scientific Procedures) Act 1986 (PPL: 70/8935). All surgery was performed under isoflurane anesthesia, and every effort was made to minimize suffering.

### Decision letter and Author response
Decision letter https://doi.org/10.7554/eLife.55761.sa1
Author response https://doi.org/10.7554/eLife.55761.sa2

## Additional files

### Supplementary files
• Transparent reporting form

### Data availability
All data, analysis scripts and simulation code is available at https://github.com/ranczlab/Galloni.etal.2020 (copy archived at https://github.com/elifesciences-publications/Galloni.etal.2020).

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
