## [Decision Letter]

**Acceptance summary:**

This study in mouse elegantly demonstrates that the massive calcium spikes commonly observed in large neocortical layer 5 pyramidal cells critically depends on dendritic trunk length, and are thus a hallmark of specific layer 5 cell types. Of additional interest, the study shows that layer 5 cells have dendritic lengths that vary widely in different cortical areas such that high calcium-dependent dendritic excitability is not ubiquitous feature across neocortex.

**Decision letter after peer review:**

Thank you for submitting your article "Apical length governs computational diversity of L5 pyramidal neurons" for consideration by *eLife*. Your article has been reviewed by three peer reviewers, including Brice Bathellier as the Reviewing Editor and Reviewer #1), and the evaluation has been overseen by John Huguenard as the Senior Editor.

The reviewers have discussed the reviews with one another and the Reviewing Editor has drafted this decision to help you prepare a revised submission. Please aim to submit the revised version within three months, but we are happy to extend this timeframe if needed, especially regarding the Covid virus crisis.

Summary:

In this study, the authors first show experimentally that L5 neurons in primary vs secondary visual cortex are substantially longer, and that they are more likely to display bursting upon somatic stimulation paired with L1 stimulation. Based on this observation the authors propose that the length of the apical dendritic trunk is key for this phenomenon and show with detailed simulations that trunk length is a critical factor to observe bursting. They finally explain this phenomenon by the combination of active Na^+^ spike backpropagation in the trunk with passive filtering (which increases with length) broadening the spikes and allowing better transfer to the tuft where they trigger a Ca^2+^ spike.

I think the idea that bursting is critically related to dendritic length is interesting (to my knowledge it was not yet proposed but I am not sure), even if it has been shown already that shorter pyramidal cells (e.g. L3) do not have large Ca+ spikes. Especially if this length varies across regions.

The manuscript by Galloni, Laffere and Rancz is an interesting study in which the authors perform experiments (slice electrophysiology) alongside computational modeling, aiming to establish a link between dendritic length and the ability of cortical pyramidal neurons to support calcium plateau potentials and subsequent somatic bursting. They focus on L5 pyramidal cells in two cortical areas, namely V1 and V2m. Their main findings are: (1) PCs in V2m are shorter and less prone to burst than PCs in V1, (2) neuronal excitability is correlated with apical trunk length, (3) shorter model neurons are less prone to burst, (4) apical tuft voltage increases with apical trunk length due to wider bAPs. The authors claim that these findings challenge the notion of canonical computations implemented by cortical neurons, whereby the coincidence of a bAP with dendritic depolarization induced calcium plateau potentials and somatic bursting (Larkum et al., 1999).

While solid evidence against the theory of canonical circuits/computations would be an important finding, I have a number of concerns that dampen my enthusiasm about this work. Specifically, I found that the study suffers from ambiguity in its terminology, makes limited use of computational modeling and presents results that are not sufficiently supported by statistical analysis.

Summary of concerns:

The general approach used by the authors (combination of slice electrophysiology and computational modeling) is particularly well-suited to derive evidence against -or in favor- of the theory of canonical circuits. However, the exploration of the parameter space should be thorough enough so as to not leave any doubts in the reader's mind about the validity of the conclusions. Although the results being presented here are extremely interesting, I cannot find them compelling enough due to a variety of reasons. These fall under three main axes: (1) strength of evidence, as established by the validity of experiments and analysis, as well as the underlying reasoning behind experiments and claims (2) flaws and omissions concerning experimental design (3) issues of clarity in terminology and presentation of results. Corresponding issues are presented below, in descending order of (subjective) importance.

The manuscript by Galloni, Laffere and Rancz titled “Apical length governs computational diversity of L5 pyramidal neurons” investigates the biophysical computations performed by layer 5 pyramidal neurons located in primary (V1) and secondary (V2m) visual cortical areas. Here, using patch clamp electrophysiology in vitro, the authors demonstrate that the computations associated with dendritic Ca^2+^ potentials vary substantially, with layer 5 pyramidal neurons in V2 having reduced dendritic excitability and less burst firing. Using modelling, the authors conclude that Na^+^ channel-dependent broadening of backpropagating action potentials underlies the influence of apical length on dendritic excitability. This study is well written and investigates an interesting phenomena which may have implications for all cortical regions. However, overall the manuscript is lacking in detail and surprisingly, there is no discussion in either the Introduction or Discussion regarding their role in the processing of visual information, which may help to put the findings in context.

Essential revisions:

Overall, combining in vivo with in silico experiments is an inspired approach that is unfortunately not seen very often. The authors have made a solid attempt to combine the two and address a very interesting research question. Unfortunately, the work suffers from several issues that detract from its validity, as described below. If these issues are successfully addressed, the work and its findings would make a significant contribution to the field.

1) The validity of the model analysis (statistical and otherwise) is not convincingly documented in several places (see below). This needs to be improved.

a) In subsection “BAC firing is absent in short ttL5 models”, the authors conclude that Ca^2+^ hotspot size does not affect excitability. However, inspection of the relevant figure (Figure 3—figure supplement 2) shows that, when comparing the traces for the 100 μm and 200 μm hotspot sizes, an additional AP is generated in the latter case, and the area under the red trace appears to be greater. This observation seems to contradict the conclusion -although it may be an outlier. The authors should perform statistical testing to support this claim, or if hot spot size plays a significant but small role compared to dendritic length, plot directly the effect of each parameter across physiological range.

Also, in this case, ADP integral measurements are not performed, and no explanation is provided to justify the differences between the two conditions (100 μm / 200 μm Ca^2+^ hotspot size).

b) In subsection “Active propagation enhances voltage in long dendrites”, it is stated that voltage attenuation is independent of trunk length for passive dendrites. However, inspecting the relevant figure (Figure 4—figure supplement 1), the top plot of subfigure c reveals that there seems to be a ~5 mV difference between the curves for the 200 and 300 μm trunk lengths as measured at the 200um mark. To justify the claim that trunk length does not impact voltage attenuation, the authors should use statistical tests to demonstrate the claimed lack of significant differences. Relying on visual inspection is not convincing.

c) In the same section, Ih reduction is stated to have no effect on the trunk length dependence of excitability (measured through bAP voltage, width and integral). However, visual inspection of the figure (Figure 4—figure supplement 3) indicates visible differences (e.g. subfigure b, width plot, tuft Ih change). In addition, prior work by Larkum et al., 2009, appears to be in contradiction with the authors' claim that Ih reduction has no effect on the trunk length dependence of excitability. As such, the claim should be either toned down or additional metrics that quantify the degree of change and deem it statistically significant should be provided.

d) In subsection “Thick-tufted L5 neuron in V2m lack a critical frequency ADP”, a claim is made concerning the properties of ttL5 neurons – but Figure 2—figure supplement 3 only provides a representative trace. A summary statistic or some other indicator that this behavior is found in all (or most) ttL5 PCs should be included.

e) In subsection “Active propagation enhances voltage in long dendrites”, it is stated that "burstiness always increased with trunk length regardless of Ra". However, no data is shown to support this – an additional figure or statistic would be useful.

f) The demonstration made in Figure 5 is not really strong. It would have been probably more convincing to vary Cm in the reduced model in the presence of back propagating APs, to show that it is local filtering of AP waveforms that allow larger spike induced depolarisation in the tuft.

2) The use of computational modeling in this study is sub-optimal. While the authors use both detailed biophysical and reduced models to support their findings, the particular model analysis design suffers from multiple flaws:

a) In general, the authors did not explore the parameter space of their models to check whether the shorter PCs would be able to generate dendritic calcium spikes and drive somatic bursting under different conditions (e.g. different inter-stimulus intervals between somatic and dendritic signals or different stimulus strength etc.). Canonical cortical computations are not synonymous to "all cells act the same" but describe a phenomenon which could be seen in many different cells, not necessarily under the exact same stimulus conditions. Given that the authors use detailed biophysical models, such an exploration is both feasible and necessary. Moreover, any interesting predictions are very easy to test in vitro, and should be tested experimentally by the authors to confirm or refute them.

b) In subsection “BAC firing is absent in short ttL5 models”, the authors state that a reduced model was selected in order to "vary dendritic length across a continuous range of values". However, it is not explained why this was necessary, as the morphology of the detailed cell of Hay et al. could also have been altered in the required manner to produce a cell with an apical trunk of the desired length. The choice to move from a detailed to a reduced model requires additional explanation, particularly when looking for a "mechanistic understanding", which is less likely to be generated by a reduced model (subsection “Active propagation enhances voltage in long dendrites”).

c) The authors report that "increasing the length of the apical trunk facilitated dendritic Ca^2+^ plateau initiation" in a reduced model, without explaining why this happens. Several possibilities exist: is it because the diameter is becoming smaller, therefore impedance is higher? Is it because the effective conductance is higher (more channels, assuming a fixed density)? Is attenuation smaller? There could be many potential explanations and the authors do not explore these possibilities in their model. Such an exploration is necessary.

d) In subsection “BAC firing is absent in short ttL5 models”, the authors describe how they altered the morphology of the detailed cell model, with the new morphology featuring a shorter apical trunk noted as the most relevant change. However, such a drastic change in morphology should have been followed by model validation (as a sanity check) to ensure basic properties remain physiological. This would also partially control for morphological changes unrelated to the length of the apical trunk. Unfortunately, no validation seems to have taken place (Materials and methods).

3) The paper is unfortunately sometimes unclear in its use of terminology, making it hard for the reader to follow all arguments to their conclusion. In addition, figures could be substantially improved, including more information (i.e. statistical) in a more efficient and effective manner. Besides important details listed in the minor points, two major clarity issues should be thoroughly addressed:

a) One of the core conclusions of this work uses a specific definition of "supralinear". In addition, a different definition is used for the same term ("supralinearities") later on (definition in Figure 2). Neither of these definitions are in line with what is commonly understood by "supralinear" in the field (i.e. a response exceeding the linear extrapolation, as seen in e.g. Branco and Hausser, 2011, Figure 1B). A common, consistent definition should be used – or a new term should be coined for the phenomenon being described.

b) The term "excitability" is sometimes used an ambiguous fashion. In paragraph two of subsection “Thick-tufted L5 neurons in V2m lack BAC firing”, "excitability" seems to be derived through the propensity of a neuron to produce bursts. However, in subsection “Thick-tufted L5 neuron in V2m lack a critical frequency ADP”, "excitability" seems to be derived through a metric dependent on the integral of the ADP, not propensity to burst. Furthermore, although the text mentions "excitability", the title of Figure 1 makes claims on bursting alone. Similar to the case in (b), a common, consistent definition should be used – or a new term should be coined for the phenomenon being described.

4) The homogeneity of the experimental data set is a concern. The recordings were performed from two-well-defined groups identified either by their projection target or genetic label. It would have been good if the conclusions could be further strengthened if further analysis was performed on these cell types. Furthermore, it is merely stated that the experiments are performed in two different projection neurons with no detail about the Glt25d2-Cre mice used, and why these particular cells were targeted in the first place. Can the authors please also illustrate which data was obtained under which conditions (ie retrograde labelling VS Glt25d2 mouse; low vs high Ca^2+^ ACSF)

5) In experiments, the supralinearity or its absence is not always thoroughly established. For example, what happens as you increase the intensity of the stimulus in Layer 1? Or the somatic current pulse? Is the response ever “linear” in either V1 or V2m neurons? I think it's important to first establish the linear regime before being able to classify a response as supralinear.

6) Overall, the manuscript is severely lacking in details associated with the findings.

Recordings were performed from layer 5 neurons with V1 and V2m. There are no details in the manuscript regarding how these brain regions were targeted in all the experiments presented. Please provide details in the Materials and methods as well as illustrations of the targeted brain regions. Are there other biophysical properties which are different between the layer 5 neurons in V1 and V2m which may explain the difference in excitability? RMP? Rheobase?

7) The experiments were performed in ACSF with 1.5 or 2 mM CaCl_2_. These recordings were pooled together as there was no significant difference between the ADP integral in the different conditions. However, p = 0.122 in V1, and observing the spread of the data points, it appears as though more experiments may indeed separate the data according to ACSF. This is not surprising, considering the authors conclude that V1 has Ca^2+^-dependent supralinearity therefore altering external Ca^2+^ concentration could be expected to alter Ca^2+^-dependent voltage in these cells.

8) During coincident input (Figure 1A), was an ADP measured in V1 and not V2 neurons? Otherwise the increased action potentials cannot be labelled as BAC firing as have no proof of calcium-dependence. V2m “usually” did not have a change in ADP. Please quantify. Also, was there a difference in the critical frequency in cells in V1 and V2m (the reported critical frequency is currently combined)?

[Editors' note: further revisions were suggested prior to acceptance, as described below.]

Thank you for resubmitting your article "Apical length governs computational diversity of layer 5 pyramidal neurons" for consideration by *eLife*. Your article has been reviewed by one peer reviewer, and the evaluation has been overseen by a Reviewing Editor and John Huguenard as the Senior Editor.

All reviewers agree that the revision has substantially improved the manuscript, however, a few points still need to be adequately addressed. In particular, the authors should make sure they sufficiently explore the parameter realm of single cell models.

Summary:

The authors have successfully addressed the vast majority of my concerns, further increasing the quality of their already impressive work. However, a few points still require clarification, and some experiments need extra support or argumentation in order to be fully convincing.

Revisions:

1) "Propensity for bursting was quantified by measuring the maximal burst ratio for each cell (defined as the largest ratio of consecutive ISIs in any current step), as well as the number of spikes in the burst and the AHP immediately following the last spike in the burst."

The "burst ratio" metric is not a widely-used metric to quantify bursting propensity, and is inadequately explained here (high value = high propensity? Why?). Please explain this better.

2) From rebuttal:

"Regarding the hot zone sizes chosen, 200 µm is the original size in Hay et al., 2011. We scaled this to the ratio of apical trunk length between the long and short morphology to get 100 µm. We added an explanation to lines subsection “BAC firing is absent in models of ttL5 neurons with short morphology”."

"The high density (“hot”) Ca^2+^ channel hotspot zone was shortened to 100 µm to reflect the 50% reduction in apical dendrite length and repositioned around the new apical branch point (350-450 µm from the soma vs 685-885 µm in the long morphology)."

It is assumed here that Ca^2+^ channel hotspot size scales linearly with apical trunk length (assuming that "dendrite" stands in for "trunk" here), which is not necessarily the case. A sub-linear or supra-linear scaling would also need to be checked. The authors appear to have actually done this (new Figure 3—figure supplement 2C/F), but do not refer to it to reinforce their point.

3) "Principled alteration of a continuous morphological parameter, such as apical trunk length, is intractable in complex dendritic morphologies."

Considering the fact that the authors are working with single-cell models, this is an inadequate justification. It is true that excessive parameterization of microcircuit models containing multiple interconnected biophysically accurate and morphologically correct reconstructions rapidly devolves into the realm of computational intractability – however, a single cell can be modeled with much greater ease – even if multiple repetitions (i.e. thousands) are required to attain a substantial sample size. If performing the full series of alterations is practically impossible due to some limitation (e.g. insufficiently powerful hardware), then it would also be adequate to show how, for a limited (but at least somewhat representative) subset of alterations, the key results obtained from a reduced morphology accurately reflect the results produced by a detailed reconstruction.

4) From rebuttal:

"We have now clarified this statement by referring to "the length dependence of Ca^2+^ electrogenesis" instead of excitability in general, and explicitly state the effect on voltage propagation (subsection “Active propagation enhances voltage in long dendrites”, and the corresponding figure legend)."

"Reducing Ih (i.e. gHCN) either in the tuft alone, or both in the trunk and tuft, while affecting voltage propagation, had no effect on the length dependence of Ca^2+^ electrogenesis"

Evidence in support of this claim is shown in Figure 4—figure supplement 3 – however, the claim is insufficiently substantiated. It is true that altering gHCN has an effect on voltage propagation, and as such, the peak voltage is also changed – which explains the change in the y-position of the curves. However, what is not shown is that the change "had no effect on the length dependence of Ca^2+^ electrogenesis" – although a visual inspection shows that all curves appear to exhibit a sudden "jump" in peak voltage (which is the result of Ca^2+^ electrogenesis, as it's absent when Ca^2+^ channels are absent), we cannot rely on mere visual inspection to ascertain whether the length at which this "jump" takes place is the same for every condition. The precise critical length should be reported for each condition, and if there are slight fluctuations, it should be demonstrated that they are not statistically significant (e.g. by repeating the experiment multiple times and taking the average of the critical length plus/minus a standard error, then showing no significant differences via statistical testing).

---

## [Author Response]

Essential revisions:Overall, combining in vivo with in silico experiments is an inspired approach that is unfortunately not seen very often. The authors have made a solid attempt to combine the two and address a very interesting research question. Unfortunately, the work suffers from several issues that detract from its validity, as described below. If these issues are successfully addressed, the work and its findings would make a significant contribution to the field.1) The validity of the model analysis (statistical and otherwise) is not convincingly documented in several places (see below). This needs to be improved.a) In subsection “BAC firing is absent in short ttL5 models”, the authors conclude that Ca^2+^ hotspot size does not affect excitability. However, inspection of the relevant figure (Figure 3—figure supplement 2) shows that, when comparing the traces for the 100 μm and 200 μm hotspot sizes, an additional AP is generated in the latter case, and the area under the red trace appears to be greater. This observation seems to contradict the conclusion -although it may be an outlier. The authors should perform statistical testing to support this claim, or if hot spot size plays a significant but small role compared to dendritic length, plot directly the effect of each parameter across physiological range.Also, in this case, ADP integral measurements are not performed, and no explanation is provided to justify the differences between the two conditions (100 μm / 200 μm Ca^2+^ hotspot size).

We have now carried out comprehensive modeling experiments varying the length of the hot zone in both morphologies between 50 and 200 µm. We have used the voltage integral at the bifurcation and the somatic ADP as output measures for the presence or absence of dendritic Ca^2+^ plateaus. No matter the length of the hot zone, the short morphology never shows Ca^2+^ plateaus, while the long morphology always does (Figure 3—figure supplement 2). The short hot zone does result in one fewer AP, however BAC firing, the target of our inquiry, is still present.

We present these findings in the main text, an extended Figure 3—figure supplement 2. Regarding the hot zone sizes chosen, 200 µm is the original size in Hay et al., 2011. We scaled this to the ratio of apical trunk length between the long and short morphology to get 100 µm. We added an explanation to subsection “BAC firing is absent in models of ttL5 neurons with short morphology”.

b) In subsection “Active propagation enhances voltage in long dendrites”, it is stated that voltage attenuation is independent of trunk length for passive dendrites. However, inspecting the relevant figure (Figure 4—figure supplement 1), the top plot of subfigure c reveals that there seems to be a ~5 mV difference between the curves for the 200 and 300 μm trunk lengths as measured at the 200um mark. To justify the claim that trunk length does not impact voltage attenuation, the authors should use statistical tests to demonstrate the claimed lack of significant differences. Relying on visual inspection is not convincing.

Indeed, there is a slight difference in the last segment due to the end effect. We have now added quantification of the length constant to Figure 4—figure supplement 1 and show statistically that it is independent on dendrite length. We have clarified now this in the Results section and the corresponding figure legend.

c) In the same section, Ih reduction is stated to have no effect on the trunk length dependence of excitability (measured through bAP voltage, width and integral). However, visual inspection of the figure (Figure 4—figure supplement 3) indicates visible differences (e.g. subfigure b, width plot, tuft Ih change). In addition, prior work by Larkum et al., 2009, appears to be in contradiction with the authors' claim that Ih reduction has no effect on the trunk length dependence of excitability. As such, the claim should be either toned down or additional metrics that quantify the degree of change and deem it statistically significant should be provided.

We have now clarified this statement by referring to “the length dependence of Ca^2+^ electrogenesis” instead of excitability in general, and explicitly state the effect on voltage propagation (subsection “BAC firing is absent in models of ttL5 neurons with short morphology” and the corresponding figure legend).

d) In subsection “Thick-tufted L5 neuron in V2m lack a critical frequency ADP”, a claim is made concerning the properties of ttL5 neurons – but Figure 2—figure supplement 3 only provides a representative trace. A summary statistic or some other indicator that this behavior is found in all (or most) ttL5 PCs should be included.

We have now characterized intrinsic properties of V1 and V2m neurons. We present this data in Figure 2—figure supplement 3 and discuss in the main text.

e) In subsection “Active propagation enhances voltage in long dendrites”, it is stated that "burstiness always increased with trunk length regardless of Ra". However, no data is shown to support this – an additional figure or statistic would be useful.

We have clarified this sentence by removing the reference to “burstiness”. We have also changed the layout of Figure 4—figure supplement 4 to show these differences more clearly.

f) The demonstration made in Figure 5 is not really strong. It would have been probably more convincing to vary Cm in the reduced model in the presence of back propagating APs, to show that it is local filtering of AP waveforms that allow larger spike induced depolarisation in the tuft.

The bAP amplitude and width are not independent and are determined by active processes along the apical trunk. We used voltage clamp to decorrelate these properties and explore the distinct contribution of the peak and width of the distal trunk voltage to tuft depolarization. We have added a schematic, changed the layout and updated the title of Figure 5.

2) The use of computational modeling in this study is sub-optimal. While the authors use both detailed biophysical and reduced models to support their findings, the particular model analysis design suffers from multiple flaws:a) In general, the authors did not explore the parameter space of their models to check whether the shorter PCs would be able to generate dendritic calcium spikes and drive somatic bursting under different conditions (e.g. different inter-stimulus intervals between somatic and dendritic signals or different stimulus strength etc.). Canonical cortical computations are not synonymous to "all cells act the same" but describe a phenomenon which could be seen in many different cells, not necessarily under the exact same stimulus conditions. Given that the authors use detailed biophysical models, such an exploration is both feasible and necessary. Moreover, any interesting predictions are very easy to test in vitro, and should be tested experimentally by the authors to confirm or refute them.

We have now explored the effect of a wide range of different inter-stimulus timings and stimulus strengths on dendritic Ca^2+^ electrogenesis in both morphologies and in different lengths of the reduced model. The conclusions remain the same: the short morphology and shorter reduced models fail to produce BAC firing, regardless of the stimulus parameters. This is now included in the main text and in Figure 3—figure supplement 1B and E.

b) In subsection “BAC firing is absent in short ttL5 models”, the authors state that a reduced model was selected in order to "vary dendritic length across a continuous range of values". However, it is not explained why this was necessary, as the morphology of the detailed cell of Hay et al. could also have been altered in the required manner to produce a cell with an apical trunk of the desired length. The choice to move from a detailed to a reduced model requires additional explanation, particularly when looking for a "mechanistic understanding", which is less likely to be generated by a reduced model (subsection “Active propagation enhances voltage in long dendrites”).

We have added a couple of sentences to explain the motivation behind switching to the simplified model (subsection “BAC firing is absent in models of ttL5 neurons with short morphology”).

c) The authors report that "increasing the length of the apical trunk facilitated dendritic Ca^2+^ plateau initiation" in a reduced model, without explaining why this happens. Several possibilities exist: is it because the diameter is becoming smaller, therefore impedance is higher? Is it because the effective conductance is higher (more channels, assuming a fixed density)? Is attenuation smaller? There could be many potential explanations and the authors do not explore these possibilities in their model. Such an exploration is necessary.

We have explored the contribution of Na^+^ channel density, distribution of ion channel along the trunk, HCN channel density as well as axial resistance. In the end, we conclude that it is an interplay between the increase in width and decrease in amplitude of the invading depolarization with increasing apical trunk length what enables Ca^2+^ plateau initiation (Figure 5D).

Regarding the suggested possibilities:

i) By construction the diameter of the apical compartment in the reduced model is kept constant throughout the entire trunk. Furthermore, the diameter of the apical compartment in the model is more difficult to interpret, as it also encompasses the membrane area of oblique dendrites, and it can thus not be directly related to the apical trunk diameter of real neurons.

ii) The density for a subset of channels is not fixed (see Materials and methods); we tested the effect of conductance gradients – thus changing the effective conductance in the last segment – and found no appreciable effect (Figure 4—figure supplements 2 and 3).

iii) Voltage attenuation is indeed smaller with increasing trunk length, but the amplitude in the last segment is still decreasing monotonically with trunk length (Figure 4—figure supplement 1B). Thus, changes in attenuation cannot explain the effect.

d) In subsection “BAC firing is absent in short ttL5 models”, the authors describe how they altered the morphology of the detailed cell model, with the new morphology featuring a shorter apical trunk noted as the most relevant change. However, such a drastic change in morphology should have been followed by model validation (as a sanity check) to ensure basic properties remain physiological. This would also partially control for morphological changes unrelated to the length of the apical trunk. Unfortunately, no validation seems to have taken place (Materials and methods).

We have now carried out the same analysis of intrinsic properties for all model neurons as for the recorded cells. This data is included in Figure 2—figure supplement 3 and shows that basic properties of all models are within the physiological range. We present this now in subsections “Critical frequency ADP is diminished in V2m ttL5 neurons” (detailed model) and “ BAC firing is absent in models of ttL5 neurons with short morphology” (reduced model) in the main text.

3) The paper is unfortunately sometimes unclear in its use of terminology, making it hard for the reader to follow all arguments to their conclusion. In addition, figures could be substantially improved, including more information (i.e. statistical) in a more efficient and effective manner. Besides important details listed in the minor points, two major clarity issues should be thoroughly addressed:a) One of the core conclusions of this work uses a specific definition of "supralinear". In addition, a different definition is used for the same term ("supralinearities") later on (definition in Figure 2). Neither of these definitions are in line with what is commonly understood by "supralinear" in the field (i.e. a response exceeding the linear extrapolation, as seen in e.g. Branco and Hausser, 2011, Figure 1B). A common, consistent definition should be used – or a new term should be coined for the phenomenon being described.

To reflect exactly the measure used, we have changed the naming of the two groups to “cfADP” and “no cfADP” (instead of no supralinearity) in Figure 2 and Figure 2—figure supplements 1 and 2. We have also changed the relevant text to be in line with the nomenclature used in the field.

b) The term "excitability" is sometimes used an ambiguous fashion. In paragraph two of subsection “Thick-tufted L5 neurons in V2m lack BAC firing”, "excitability" seems to be derived through the propensity of a neuron to produce bursts. However, in subsection “Thick-tufted L5 neuron in V2m lack a critical frequency ADP”, "excitability" seems to be derived through a metric dependent on the integral of the ADP, not propensity to burst. Furthermore, although the text mentions "excitability", the title of Figure 1 makes claims on bursting alone. Similar to the case in (b), a common, consistent definition should be used – or a new term should be coined for the phenomenon being described.

We have now replaced the general term of excitability with more precise ones, like propensity for bursting or dendritic Ca^2+^ electrogenesis. We have however, retained excitability where used in a broad and general term, meaning the ability to respond to input in a regenerative fashion.

4) The homogeneity of the experimental data set is a concern. The recordings were performed from two-well-defined groups identified either by their projection target or genetic label. It would have been good if the conclusions could be further strengthened if further analysis was performed on these cell types. Furthermore, it is merely stated that the experiments are performed in two different projection neurons with no detail about the Glt25d2-Cre mice used, and why these particular cells were targeted in the first place.

The target cell type of our investigation was ttL5, to which both the Glt and the retro-labelled neurons belong. Figure 1—figure supplement 1 shows there is overlap between the two subpopulations, but the extent of this overlap is immaterial as they both belong to the same cell type. We have now clarified that the Glt25d2-cre labelled neurons belong to the ttL5 cell type. An in-depth comparison of Glt and retro-labelled subpopulations is outside of the scope of this paper.

Can the authors please also illustrate which data was obtained under which conditions (ie retrograde labelling VS Glt25d2 mouse; low vs high Ca^2+^ ACSF)

We illustrate the provenance of all low calcium and Glt labelled neurons in Figure 2—figure supplements 1 and 2 and state this now in the respective figure legends.

5) In experiments, the supralinearity or its absence is not always thoroughly established. For example, what happens as you increase the intensity of the stimulus in Layer 1? Or the somatic current pulse? Is the response ever “linear” in either V1 or V2m neurons? I think it's important to first establish the linear regime before being able to classify a response as supralinear.

We acknowledge this use of the term does not follow the convention used in the field, thus we replaced it with “nonlinear” instead. When establishing supralinearity, we have considered the spike output. With this approach, V2m neurons generally responded with 1 spike to the combined stimulus of 1 somatically triggered spike + L1 synaptic input, hence they were termed linear (i.e. 1 + 0 = 1). V1 neurons on the other hand responded with 3 spikes to the combined stimulus of 1 somatically triggered spike + L1 synaptic input, hence they were termed supralinear (1 + 0 ≠ 3).

To address the question of linearity directly at the level of somatic membrane potential, we now carried out additional analysis of our BAC dataset, illustrated in Author response image 1. We have measured the somatic voltage integral in a 50 ms window after the onset of L1 or somatic stimulus. We then compared the integral measured during coincident stimulation to the arithmetic sum of integrals following each stimulus alone for both V1 (top row, n = 21) and V2m (bottom row, n = 18) neurons. To explore both linear and supralinear regimes, the measurements were performed with both low and high L1 stimulus intensities, while somatic stimulation was kept slightly above rheobase in order to consistently evoke exactly one spike. Low L1 stimulus intensities (left column) were tuned to evoke a somatic EPSP of approximately 5 mV in amplitude. In this stimulus regime, all responses summed approximately linearly. High L1 stimulus intensities (right column) were set to evoke no more than one spike at the soma. In this regime, during coincident stimulation a subset of V1 neurons (labelled in red) exhibited BAC firing in the form of larger depolarization and a burst of 3 or more spikes. This is reflected in the voltage integral exceeding the arithmetic sum significantly. The remaining neurons (black) only ever produced at most two spikes, hence were not BAC firing and the voltage integral during coincident stimulus was similar to the arithmetic sum of the two individual stimuli. In V2m, summation was always sublinear, regardless of L1 input intensity.

To summarize, the coincident stimulation only resulted in supralinear summation when BAC firing was evoked in V1. In all other cases summation was either linear or sublinear. We note however that the somatic voltage integral is a convoluted measure affected by different AHP and ADP magnitudes as well as the number of spikes. We thus decided not to include this figure as supplementary material.

**Author response image 1. sa2fig1:** 

6) Overall, the manuscript is severely lacking in details associated with the findings.Recordings were performed from layer 5 neurons with V1 and V2m. There are no details in the manuscript regarding how these brain regions were targeted in all the experiments presented. Please provide details in the Materials and methods as well as illustrations of the targeted brain regions.

Brain areas were targeted based on stereotaxic coordinates for V1 and V2m found in the Franklin and Paxinos, 2007 brain atlas, as described in the Materials and methods section. In addition, we have now substantially expanded Figure 1—figure supplement 1 to show the location of all cells recorded in the cfADP experiments, which represents the bulk of our recordings. Currently we can’t determine the exact location of the cells from the BAC firing experiments due to the Covid-19 lockdown, but the very same targeting approach was used.

Are there other biophysical properties which are different between the layer 5 neurons in V1 and V2m which may explain the difference in excitability? RMP? Rheobase?

This question is answered in major point 1d.

7) The experiments were performed in ACSF with 1.5 or 2 mM CaCl_2_. These recordings were pooled together as there was no significant difference between the ADP integral in the different conditions. However, p = 0.122 in V1, and observing the spread of the data points, it appears as though more experiments may indeed separate the data according to ACSF. This is not surprising, considering the authors conclude that V1 has Ca^2+^-dependent supralinearity therefore altering external Ca^2+^ concentration could be expected to alter Ca^2+^-dependent voltage in these cells.

Our statement relates purely to the technical ability to pool these two datasets and we do not deny the quantitative role extracellular Ca^2+^ concentration plays in the magnitude of Ca^2+^ currents. To clarify this, we have reworded the corresponding part and added the exact p values together with the test statistics to the main text. Furthermore, both the magnitude and the similarity of sample sizes across conditions and brain areas further supports the validity of pooling.

8) During coincident input (Figure 1A), was an ADP measured in V1 and not V2 neurons? Otherwise the increased action potentials cannot be labelled as BAC firing as have no proof of calcium-dependence.

We have now carried out the measurements requested and quantified the somatic voltage integral (see response to major point 5). In BAC firing (i.e. bursting) neurons this complex ADP measure shows supralinear summation, but we decided not to add this data to the supplementary material for the following reasons:

The relationship between the timing of the last somatic spike in the burst and the termination of the Ca^2+^ plateau is much more complex than in the cfADP experiments, hence an ADP is a misleading measure and is not directly comparable to the cfADP. Furthermore, the output measure for the experiments in Figure 1A was the nonlinear spike output to coincident stimulation (see response to major point 5) and not the ADP.

V2m “usually” did not have a change in ADP. Please quantify.

ADP values are quantified as shown in Figure 1C and plotted in Figure 2A.

Also, was there a difference in the critical frequency in cells in V1 and V2m (the reported critical frequency is currently combined)?

We updated the text and Figure 1E to display critical frequency data separately for V1 and V2m neurons.

[Editors' note: further revisions were suggested prior to acceptance, as described below.]

Revisions:1) "Propensity for bursting was quantified by measuring the maximal burst ratio for each cell (defined as the largest ratio of consecutive ISIs in any current step), as well as the number of spikes in the burst and the AHP immediately following the last spike in the burst."The "burst ratio" metric is not a widely-used metric to quantify bursting propensity, and is inadequately explained here (high value = high propensity? Why?). Please explain this better.

We thank the reviewer for pointing out that the explanation here was too brief. We devised this novel metric to capture the widest range of bursting behaviours. We have updated paragraph three of subsection “Critical frequency ADP is diminished in V2m ttL5 neurons” and subsection “Data acquisition and analysis”.

2) From rebuttal:"Regarding the hot zone sizes chosen, 200 µm is the original size in Hay et al., 2011. We scaled this to the ratio of apical trunk length between the long and short morphology to get 100 µm. We added an explanation to lines subsection “BAC firing is absent in models of ttL5 neurons with short morphology”.""The high density (“hot”) Ca^2+^ channel hotspot zone was shortened to 100 µm to reflect the 50% reduction in apical dendrite length and repositioned around the new apical branch point (350-450 µm from the soma vs 685-885 µm in the long morphology)."It is assumed here that Ca^2+^ channel hotspot size scales linearly with apical trunk length (assuming that "dendrite" stands in for "trunk" here), which is not necessarily the case. A sub-linear or supra-linear scaling would also need to be checked. The authors appear to have actually done this (new Figure 3—figure supplement 2C/F), but do not refer to it to reinforce their point.

We have indeed tested both sub- and supra-linear scaling of hot zone size and we agree with the reviewer that these are important controls worthy of highlighting. We now refer to them directly and replaced “dendrite” with “trunk”.

3) "Principled alteration of a continuous morphological parameter, such as apical trunk length, is intractable in complex dendritic morphologies."Considering the fact that the authors are working with single-cell models, this is an inadequate justification. It is true that excessive parameterization of microcircuit models containing multiple interconnected biophysically accurate and morphologically correct reconstructions rapidly devolves into the realm of computational intractability – however, a single cell can be modeled with much greater ease – even if multiple repetitions (i.e. thousands) are required to attain a substantial sample size. If performing the full series of alterations is practically impossible due to some limitation (e.g. insufficiently powerful hardware), then it would also be adequate to show how, for a limited (but at least somewhat representative) subset of alterations, the key results obtained from a reduced morphology accurately reflect the results produced by a detailed reconstruction.

We apologize for having been inaccurate with our phrasing and thank the reviewer for helping us make our reasoning clearer. The reviewer correctly points out that we do not face computational intractability in this instance. Instead, this is a conceptual question in our view. The morphologically detailed model runs on a real biological morphology, which inherently has a variety of co-dependent factors (such the positions of oblique dendrites) that are also known to influence bursting behaviour. While it is indeed possible to arbitrarily alter the morphology, either manually or through generative models such as the TREES toolbox, such a chimera of biological and artificial morphology would make it difficult to interpret the result and may show behaviour that is misleading. It is for this reason that we made the explicit decision when designing our study to focus this investigation on the reduced model, where such morphological alterations are less ambiguous.

We accept that this may be a conceptual point where our opinions differ. We hope to have made our reasoning clearer by rephrasing the relevant sentences.

4) From rebuttal:"We have now clarified this statement by referring to "the length dependence of Ca^2+^ electrogenesis" instead of excitability in general, and explicitly state the effect on voltage propagation (subsection “Active propagation enhances voltage in long dendrites”, and the corresponding figure legend).""Reducing Ih (i.e. gHCN) either in the tuft alone, or both in the trunk and tuft, while affecting voltage propagation, had no effect on the length dependence of Ca^2+^ electrogenesis"Evidence in support of this claim is shown in Figure 4—figure supplement 3 – however, the claim is insufficiently substantiated. It is true that altering gHCN has an effect on voltage propagation, and as such, the peak voltage is also changed – which explains the change in the y-position of the curves. However, what is not shown is that the change "had no effect on the length dependence of Ca^2+^ electrogenesis" – although a visual inspection shows that all curves appear to exhibit a sudden "jump" in peak voltage (which is the result of Ca^2+^ electrogenesis, as it's absent when Ca^2+^ channels are absent), we cannot rely on mere visual inspection to ascertain whether the length at which this "jump" takes place is the same for every condition. The precise critical length should be reported for each condition, and if there are slight fluctuations, it should be demonstrated that they are not statistically significant (e.g. by repeating the experiment multiple times and taking the average of the critical length plus/minus a standard error, then showing no significant differences via statistical testing).

Our original focus was on determining which conductance is critical to the length dependence, i.e. to the general finding that peak tuft voltage increases with trunk length. As we show, gNa is critical to this length dependence, while gHCN appear to only have a minor modulatory role. We paid less attention to modulatory effects.

We have carried out extra modelling to elaborate the modulatory effect of gHCN. We have now quantified the effect of gHCN on both the voltage propagation and critical (threshold) length. Please note that while gHCN does affect the nominal threshold length somewhat, our main finding, namely that there is length dependence of Ca^2+^ electrogenesis, persists even at zero I_h_.

The results of this analysis are now shown in Figure 4—figure supplement 3 CD. We have extended the relevant text in subsection “Active propagation enhances voltage in long dendrites”. As the analysis of threshold length required finer granularity simulations, we have also updated Figure 4B-E as well as Figure 4—figure supplement 3 AB.